



# On structural errors in emergent constraints

Benjamin M. Sanderson[1,3], Angeline Pendergrass[2,3], Charles D. Koven[4], Florent Brient[5], Ben B. B. Booth[6], Rosie A. Fisher[1,3] and Reto Knutti[7]

[1]CERFACS, Toulouse, France
[2]Cornell University, Ithaca, NY, USA
[3]National Center for Atmospheric Research, Boulder, CO, USA
[4]Lawrence Berkeley National Lab, CA, USA
[5]LMD/IPSL, Sorbonne Université, Paris, France
[6]UK Met Office, Exeter, UK
[7]ETH Zurich, Switzerland

*Correspondence to*: Benjamin Sanderson (sanderson@cerfacs.fr)

## ABSTRACT.

Studies of 'emergent constraints' have frequently proposed that a single metric alone can constrain future responses of the Earth system to anthropogenic emissions. The prevalence of this thinking has led to literature and messaging which is sometimes confusing to policymakers, with a series of studies over the last decade making confident, yet contradictory, claims on the probability bounds of key climate variables. Here, we illustrate that emergent constraints are more likely to occur where the variance across an ensemble of climate models of both observable and future climate arises from common structural assumptions and few degrees of freedom. Such cases are likely to occur when processes are represented in a common, oversimplified fashion throughout the ensemble, about which we have the least confidence in performance out of sample. We consider these issues in the context of a number of published constraints, and argue that the application of emergent constraints alone to estimate uncertainties in unknown climate responses can potentially lead to bias and overconfidence in constrained projections. Together with statistical robustness and plausibility of mechanism, assessments of climate responses must include multiple lines of evidence to identify biases that arise from common oversimplified modeling assumptions which impact both present and future climate simulations in order to mitigate against the influence of common structural biases.

## 1 INTRODUCTION

Models of the climate system face a particular challenge: their primary purpose is to project the future response of the Earth system to boundary conditions which have yet to be realized. Confidence in models' future projections cannot come from iterative verification and improvement, but instead must be grounded in a combination of an understanding of the adequacy of simulation of relevant Earth System feedback processes, together with an assessment of the degree to which the models can represent historical behaviour. The latter can potentially provide metrics or constraints that can inform which configurations of each model are most defensible as tools to project future climates.

In climate model development and calibration, these types of constraints are utilised in an extended expert assessment where biases in climatology and historical trends are iteratively reduced and addressed through improved process representation and parameter adjustment (Hourdin et al., 2017; Mauritsen et al., 2012; Schmidt et al., 2017), or systematically through the use of perturbed ensembles and formal inference (Tett et al., 2017; Williamson et al., 2013; Zhang et al., 2018). Adequate performance on a subset of metrics is generally accepted as necessary for consideration as a member of the collection of climate models (Eyring et al., 2016) used to assess future change in IPCC assessment reports (Pachauri et al., 2014) - for example, the need for models to conserve energy or to broadly reproduce the



observed global mean temperature evolution of the 20th Century. Other performance metrics may be of particular interest to specific modeling centers - for example, reducing biases in the simulation of a particular regional climate or for a particular application (for example, for simulating climate features relevant for energy infrastructure (Golaz et al., 2019) or optimizing model performance at high latitudes (Tjiputra et al., 2020)).

Recent literature (Bretherton and Caldwell, 2020; Brient, 2019; Cox, 2019; Eyring et al., 2019; Hall et al., 2019; Klein and Hall, 2015) has also focused on a class of "emergent" constraint which differs conceptually in that the relevance of the metric is defended by the existence of  correlations between a potentially observable metric and a projected future climate response, within an ensemble of ESM simulations.  Emergent constraints are generally applied in a regression framework, where the ensemble is used to define a predictive relationship which can be used, together with observations, to produce an estimate of constrained projected values.   Examples have considered constraints of Equilibrium Climate Sensitivity (hereafter ECS) from aspects of natural variability (Cox et al., 2018b) or cloud properties (Brient and Schneider, 2016; Sherwood et al., 2014), Transient Climate Response (TCR) from observed warming trends (Nijsse et al., 2020; Tokarska et al., 2020) and future carbon cycle (Cox, 2019) and ice-albedo feedbacks (Cox, 2019; Qu and Hall, 2007; Thackeray and Hall, 2019) from their observed seasonal variations.

There are a number of factors that have been recognized which might lead to overconfidence in the constrained projections arising from the use of emergent constraints.  The first is that, because of the relatively small sample size in CMIP ensembles (or small effective sample size due to model interdependencies (Sanderson et al., 2015)) and the large number of outputs, it is inevitable that some variables will be correlated with climate response metrics by chance (Caldwell et al., 2014).  This means that our confidence in a constraint cannot arise from correlation across the ensemble alone, but also from the plausibility of the proposed mechanism which relates the proposed predictor to the future climate response (Caldwell et al., 2018).  However, although many published emergent constraints propose a physical explanation for an underlying process which might jointly control the predictor and predictand, robust demonstration of a mechanism often requires tools which might not be available, such as systematic sampling of parameters and process representations in models (Hall et al., 2019; Klein and Hall, 2015)

At least some emergent constraints can be shown to be overconfident using existing data, by considering the inconsistency of constraints over different generations of model intercomparisons (Klein and Hall, 2015) or lack of agreement of different constraints on the same quantity in the literature (Brient, 2019). Such disagreement might arise due to inconsistency in the definition of a climate response: for example, if ECS is in fact dependent on climate state then the value inferred from cooling during the last glacial maximum would differ from that inferred from recent decades.   But overconfidence could also arise from overly strong statistical assumptions on the robustness of ensemble derived relationships (Williamson and Sansom, 2019). The standard regression model uses an ensemble-derived regression relationship between predictor (the potentially measurable variable) and predictand (the unknown climate response) to make a calibrated projection,  implicitly assuming the real world is *exchangeable* with models in the ensemble, which is to say that the relationship is equally likely to apply to the real world as to members of the model ensemble.

There are a number of reasons why we might not expect this assumption of exchangeability to hold.  We know that the models which populate our ensembles are subject to limits of resolution and complexity. This means that they can be considered only as approximations of the real world, likely with more in common with each other than reality (an issue which can be compounded by replicated assumptions and components within the ensemble (Caldwell et al., 2014; Sanderson et al., 2015)).  Even in the presence of a strong correlation and a plausible physical mechanism explaining the constraint in simulations (Caldwell et al., 2018), the correlation might only arise due to common simplifications throughout the ensemble.  Such concerns have led to debate as to whether emergent constraints should be included in integrative assessments of uncertainty in ECS (Sherwood et al., 2020), underlining the need for a robust framework in which to consider emergent constraints as lines of evidence.



A first step towards more robust use of emergent constraints is to combine different lines of evidence (Bretherton and Caldwell, 2020; Brient, 2019), effectively relaxing the assumption that a single constraint is reliable (but maintaining that constraints have some potential value, even if they disagree). However, enacting this approach requires considering a number of additional factors: the degree to which each component constraint has a plausible mechanism (Caldwell et al., 2018) and the degree of independence assumed between different constraints (Bretherton and Caldwell, 2020).

Uncertainties in the relationship and in the source ensemble can at least be represented by framing the problem in a Bayesian framework (Hargreaves et al., 2012; Renoult et al., 2020) or using information theory approaches (Brient and Schneider, 2016). These frameworks can naturally allow the integration of multiple constraints by effectively weighting the climate responses of different models in the ensemble by likelihood informed by a set of constraints. Critically, they can also be expanded to represent the likelihood that ensemble members are exchangeable with reality (Williamson and Sansom, 2019) (which is effectively assumed in most studies published to date). But even in an ideal case, elements of the calibration of the statistical model parameters would remain somewhat subjective, conditional on prior assumptions about climate responses and chosen metrics of model adequacy and interdependency.

Figure 1 shows a collection of published emergent constraints on a commonly studied predictand, ECS, for CMIP5 (see Table 1), together with a simple application of each single constraint using a common analysis. In each case, given an input vector of constraints and corresponding values of ECS for CMIP models (see Table 3), uncertainty in the regression relationship is estimated using a bootstrap approach. 1000 bootstrap samples of models are created, with replacement. The result is 1000 estimates of the best-fit line, illustrated by red lines on the diagonal plots on Figure 1. Observed maximum and minimum values are considered to correspond to the 10th and 90th percentile of an 'observational' normal distribution (illustrated by grey rectangles in Figure 1). Constrained distributions are then created by drawing 1000 members from the observational distribution, and using each member of the bootstrap regression estimate in turn to produce a series of estimates of ECS, illustrated by green rectangles on the diagonal plots in Figure 1. Models with above and below median climate sensitivity are shown in red and blue, respectively.

Pairwise combinations of constraints are illustrated on the off-diagonal plots, with ellipses illustrating the 10th and 90th percentiles of the corresponding observational range on the major and minor axes of the ellipse (an illustration, not used in further calculation, which assumes that the constraints are independent, which it should be noted may not necessarily be the case (Bretherton and Caldwell, 2020)).

In some cases, predictors are not well correlated with each other, and the combination of predictors appears to increase the explained variance in the net feedback relative to either constraint alone (Caldwell et al., 2018) (see for example, the pair of Sherwood D and S constraints, or the pair of Tian (2015) and Qu et (2014) constraints). In these cases, combining different sources of information in a classical (Bretherton and Caldwell, 2020) or Bayesian (Brient, 2019) framework might be appropriate, given appropriate checks for plausibility of mechanisms (Caldwell et al., 2018; Hall et al., 2019) and robustness of sample (Caldwell et al., 2014) . Though as we discuss in the following section, a predictive model for the set of feedbacks which vary within the ensemble may still be overconfident when applied out of sample if key processes are generally missing or oversimplified in the ensemble.



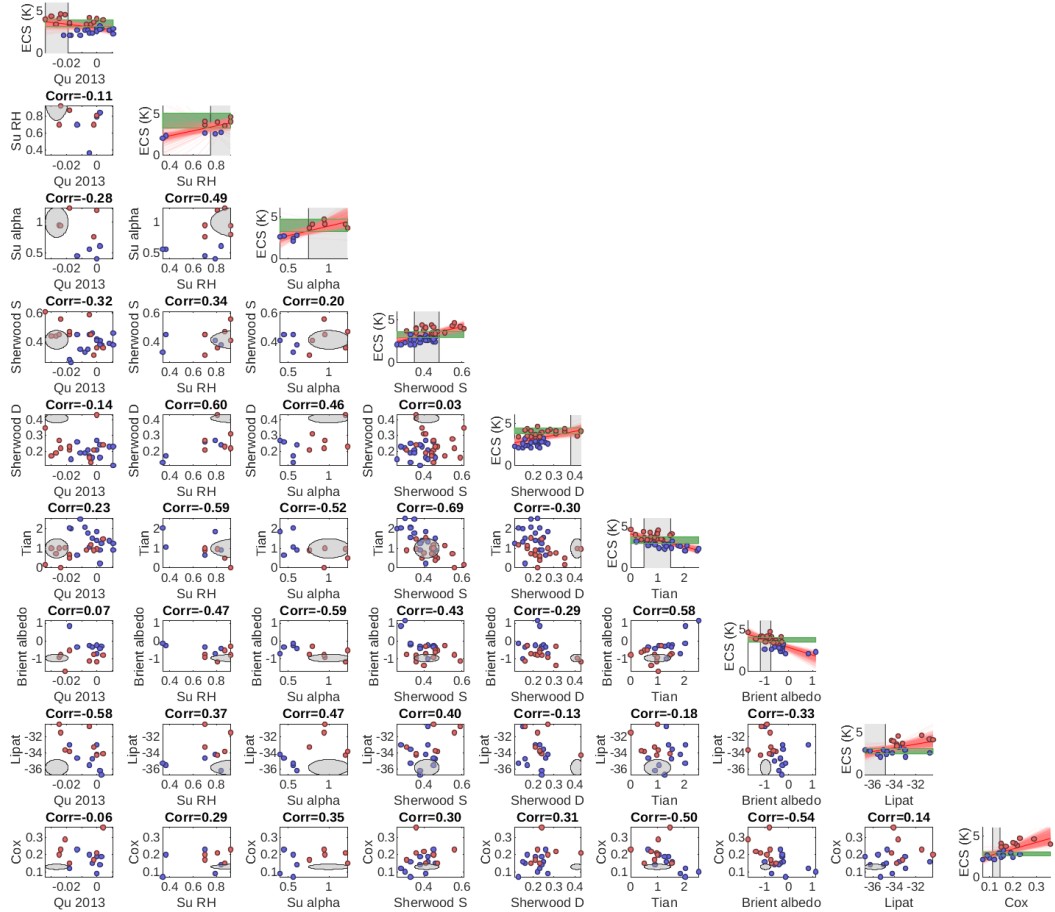

**Figure 1. Comparison of a selection of emergent constraints on equilibrium climate sensitivity (detailed individually in Table 1).** Off-diagonal plots show the pairwise comparison of each emergent constraint with all others. Ellipse major and minor axes illustrate the observational ranges proposed in the original study for each constraint. Diagonal plots show the published constraint on the x-axis, and ECS on the y axis. Blue and red points represent models with values of ECS above and below the ensemble median, respectively. The vertical gray bar shows the observed range as reported in the original papers (Brient and Schneider, 2016; Cox et al., 2018b; Lipat et al., 2017; Qu et al., 2014; Sherwood et al., 2014; Su et al., 2014; Tian, 2015). The red lines show possible linear fits considered in a 1000 member bootstrap regression. Shaded green rectangles show the 10th and 90th percentiles of inferred distribution of ECS using the bootstrap regression estimate.





| Emergent Constraint | Reference | Proposed Metric |
|---|---|---|
| Qu | (Qu et al., 2014) | Pre-industrial gradient of low cloud cover as a function of Sea Surface Temperature (1/K) |
| Klein | (Klein et al., 2013) | E(CTP/tau) - error in cloud fraction integrated over ISCCP cloud top pressure/optical thickness bins |
| Su RH | (Su et al., 2014) | Scaling of low-level relative humidity from AIRS between 45S to 40N |
| Su alpha | (Su et al., 2014) | Scaling zonal mean cloud fraction from CloudSat/CALYPSO between 45S to 40N |
| Sherwood S | (Sherwood et al., 2014) | Index of lower tropospheric convective mixing |
| Sherwood D | (Sherwood et al., 2014) | Index of large scale lower tropospheric mixing |
| Brient Shal | (Brient et al., 2016) | Fraction of marine tropical boundary layer cloud fraction below 950mb |
| Zhai | (Zhai et al., 2015a) | Boundary layer cloud fraction response to seasonal temperature variation |
| Tian | (Tian, 2015) | Scaling of simulated precipitation in ICTZ relative to GPCP observed values |
| Brient albedo | (Brient and Schneider, 2016) | Gradient of SW cloud reflectivity as a function of tropical SST |
| Lipat | (Lipat et al., 2017) | Edge latitude of Hadley cell |
| Cox | (Cox et al., 2018b) | Psi - fluctuation/dissipation metric, function of global mean temperature variance and lag-covariance |

**Table 1. Emergent constraint metrics on equilibrium climate sensitivity used in Figure 1.**

In other cases, pairs of predictors are well correlated with each other (making it unlikely that they describe different processes), but the resulting observational constraint on real-world ECS is inconsistent. For example - the Cox variability constraint (Cox et al., 2018b) predicts a relatively low value for ECS and the Sherwood "D" constraint from the ratio of shallow to deep overturning (Sherwood et al., 2014) predicts a high value (one could also consider Lipat et al. (2017) and Qu et al. (2014)). In each case, the constraints are correlated with each other (indeed, both constraints have been found to correlate with similar feedback processes (Caldwell et al., 2018)). However, here we observe that the two constraints disagree on the implications for ECS because observations in the 2D space defined by the two constraints fall outside the ensemble distribution. Such disagreement implies that one or both of the constraints is overconfident, but this could occur for a number of reasons:

1. Two constraints might disagree because one (or both) are spurious, arising from insufficient samples in the ensemble.
2. Errors in observations of either quantity may lead to an erroneous constraint and apparent disagreement.
3. One or both of the constraints could arise due to structural deficiencies in how processes are represented in the model - a predictor-predictand relationship could exist within the common simplified framework of model parameterizations, but it does not apply to the real-world (and this is manifested by the disagreement of constraints)

To understand the latter case better, we can consider a situation where we know that our ensemble explores only a single model structure which is oversimplified compared to the real world.



## 2 A LESSON FROM PARAMETER PERTURBATION EXPERIMENTS


Although the concept of emergent constraints as applied to multi-model ensembles has become popular in the last decade, the general formulation was used previously in the perturbed parameter literature. (Piani et al., 2005) used a statistical formulation which might today be classified as an emergent constraint, identifying statistical modes of variability which were correlated with climate sensitivity in a large ensemble of perturbed parameter experiments

(PPEs), then using observations to produce constrained estimates of ECS. The ensemble used in this case was sufficiently large (Stainforth et al., 2005) that the relationships were statistically robust in sample, but were found to be inaccurate when applied to an out of sample set of simulations (in this case, predicting the climate sensitivity of members of the CMIP ensemble (Sanderson, 2013)).

To understand why this is the case, we must consider the conceptual differences between perturbed-parameter and

multi-model ensembles. In PPEs, a single model structure is used, and both predictors and predictands are functions of the parameters which are perturbed in the experiment. Emergent constraints in a PPE are generally easy to find (Knutti et al., 2006; Piani et al., 2005; Sanderson, 2011; Yokohata et al., 2010) because there is a low-dimensional functional relationship between predictors and future response in the ensemble - both are, by construction, functions of the perturbed input parameters. Feedback variation in a PPE is a function of a subset of the parameters which have

been perturbed; thus, if any potentially observable quantities are also functions of those same parameters, an emergent constraint is automatically present. Due to this underlying parametric structure, many emergent constraints can be found in a PPE; but they are not individually useful, because there are no model versions which satisfy all constraints simultaneously due to the structural component of model error which cannot be tuned (Sanderson et al., 2008), and their predictions are generally not applicable to other models (Sanderson, 2011, 2013; Yokohata et al., 2010) (an effect

which has been observed in multi-structure PPEs (Kamae et al., 2016)).

In model calibration exercises, structural errors in a single model manifest through differences in optimal parameter configurations which arise from prioritizing different observations in the cost function. For example, different optimal parameter configurations minimize errors in the Amazon and Indonesian rainforests (McNeall et al., 2016), implying an underlying structural error in the model which requires that a global calibration must be a trade-off in biases in the

two regions, leaving an irreducible error which cannot be eliminated by parameter adjustment alone.

It is understood that probabilistic predictions of future changes made from a PPE must be robust in the face of this irreducible error (Rougier, 2007). In some cases, the MME has been used as an out of sample test to assess overconfidence in predictions made from relationships within the PPE (Sanderson, 2013; Sexton and Murphy, 2012). The correspondence between model errors and the model parameter space also allows for the conceptualization and

quantification of error trade-offs through 'history matching' (McNeall et al., 2016; Williamson et al., 2013) (approaches which rule out parts of parameter space that perform poorly in multiple metrics). Such approaches can retain a subset of model variants with comparable net errors but with different tradeoffs (in the simple example above, including model versions which minimize errors in either the Amazon or Indonesian rainforests).

Such strategies seek to incorporate model performance in reproducing a range of observables using a model which is

imperfect, where it is understood that placing all emphasis on a single observable (as in an emergent constraint) would lead to overconfidence. In a PPE, this is demonstrable because we have a wider structural sample (the MME) in which predictions can be tested, and because model errors can be represented as a function of model parameters which helps us both conceptualize and quantify systematic errors.

In an MME, we do not have similar out-of-sample estimates to illustrate the limitations of ensemble-derived

correlations, and there is not necessarily a simple underlying parametric structure which allows us to quantify how assumptions map onto errors. But, this does not imply that the literal interpretation of emergent constraints is justified. Our experience with PPEs has shown that emergent constraints can arise due to an underlying parametric structure - which is present by construction in a PPE, but may also be effectively present in an MME if the same parameterizations



are used throughout the ensemble. This is a potential source of overconfidence in existing ECs which is not generally
accounted for.

If an MME includes subsets of models with common structural assumptions, it is also possible that ECs may exist
within a given subset. In such cases, confidence in the emergent constraint should be conditioned on the degree to
which the models in the subset are plausible. Underlying these uncertainties is a requirement for independently
assessing the likelihood or plausibility of model structures.

In short, we cannot easily quantify the impacts of structural error in MME-derived ECs, but equally, it is not justifiable
to assume that the MME is interchangeable with reality or that common structural errors are absent. Indeed, the very
presence of an EC for a given process in an MME might be indicative of a lack of diversity of process representation
because constraints are more likely to emerge if there are limited effective degrees of freedom represented in the
ensemble. Robust multi-metric approaches which are a demonstrable necessity in a PPE are equally advisable in an
MME.

## 3 THE NATURE OF MULTI-MODEL EMERGENT CONSTRAINTS

How then do we assess whether an ensemble is sufficiently structurally diverse that an emergent constraint arising
from it could be applicable to the real world? In a PPE, constraints can be tested to some extent by testing
relationships in the MME, which we can assume contains a larger structural sample; but for an MME, we have no
such superset. If an emergent constraint has been found in an MME (providing it has been demonstrated not to be
statistically spurious by, for example, persisting through multiple generations of ensemble (Bracegirdle and
Stephenson, 2013)), what remains is to assess the degree to which that emergent constraint can be applied to reality
(Williamson and Sansom, 2019).

Here, we propose that ECs can be categorised conceptually, and by doing so, the nature of their potential structural
errors can be better evaluated. We consider three 'kinds' of EC:

### 3.1 CONSTRAINTS OF THE FIRST KIND: BIAS PERSISTENCE/SIGNAL EMERGENCE

The first kind of constraint includes cases where the measured quantity and the unknown quantity are of the same
nature, such that both are expressions of a system's response to a forcing with comparable spatial and temporal
features. For example, if the observed historical warming in an MME is used to constrain the warming in a future
scenario (Jiménez-de-la-Cuesta and Mauritsen, 2019), both predictor and predictand are expressions of global mean
warming in response to a gradually increasing greenhouse gas forcing (constraining Transient Climate Response
through observed warming (Nijsse et al., n.d.) could be argued to fall into this category). Other examples include the
conditioning of future sea-ice extent trends on historical trends (Boé et al., 2009; Knutti et al., 2017), constraining the
range of future soil moisture with its observed transient historical trends (Douville and Plazzotta, 2017) and the
persistence of carbon dioxide concentration biases in emissions-driven simulations (Hoffman et al., 2014)

These examples all broadly concern an emergent transient signal in response to a gradual increase in anthropogenic
forcing over time, so they are effectively statements that a bias in transient response is likely to persist if forcing
continues to increase. Because these constraints directly measure the trend itself, they are relatively insensitive to
model assumptions in how and why a trend is simulated, provided there exists a robust relationship between the given
aspect of future behaviour and its historical trend.

This assumption is valid if it can be defended that both predictor and projected quantity are describable as functions
of the same emerging trend. The resulting EC is effectively a (potentially nonlinear) extrapolation, where the strength
of the relationship is conditional on the degree to which models represent similar nonlinearities. The relationship is
not strongly conditional on underlying structural assumptions because biases are manifested similarly in the historical
and future trends. The strength of the correlation in the EC reflects the degree to which models agree on the form of



the extrapolation, and thus the only concern for overconfidence is if the relationship between past and future trends is similarly biased in many models (through the common omission of a state-dependent nonlinearity, for example, or a missing forcing in one period in most models).

Constraints of this type are similar to the classical detection problem (Hegerl and Zwiers, 2011; Ribes et al., 2017) where the amplitude of an emerging signal in response to a forcing is estimated in the presence of noise arising from internal variability and other confounding forcers. There exists a large literature in performing such detection of a signal response to a forcing in the context of noise, model errors and other confounding forcings (Hegerl and Zwiers, 2011).

### 3.2 CONSTRAINTS OF THE SECOND KIND: FEEDBACK PROCESS ISOLATION

The second kind of EC involves the identification of a primary feedback mechanism which governs the future response, and the subsequent proposal of an observable quantity which constrains the strength of that feedback within the ensemble. There are a large number of ECs which fall into this category for ECS (Brient et al., 2016; Lipat et al., 2017; Sherwood et al., 2014; Siler et al., 2018; Su et al., 2014; Tian, 2015; Trenberth and Fasullo, 2010; Volodin, 2008; Zhai et al., 2015b), in most cases involving mechanistic constraints on the response of shallow convective clouds to warming (considered to be the primary source of uncertainty in ECS in CMIP5 (Andrews et al., 2012) and CMIP6 (Zelinka et al., 2020) ). Other studies propose to directly constrain individual cloud feedbacks (Brient et al., 2016; Gordon and Klein, 2014; Qu et al., 2014; Siler et al., 2018) or future precipitation changes (Allen and Ingram, 2002; Watanabe et al., 2018).

Emergent constraints obtained by statistical data-mining (either transparently or otherwise) (Caldwell et al., 2014) can potentially fit into this category, though in order to be defensible, such constraints must be demonstrated to be statistically robust (Caldwell et al., 2014) and also provide a plausible mechanism to explain why the candidate process is the dominant factor in explaining ensemble variance in future response, and why the proposed observable is an expected metric of that process (Caldwell et al., 2018; Hall et al., 2019).

However, unlike constraints of the first kind, a process-based constraint does not describe uncertainty in future response in a general sense - at best, it describes the leading order process which explains variability in future response across the ensemble. A plausible, robust, process-based EC is still conditional on the plausibility of the relevant process as it is represented in the class of models used in the ensemble.

### 3.3 CONSTRAINTS OF THE THIRD KIND: FREQUENCY SUBSTITUTION

The third kind of constraint proposes that the future response to a given forcing **A** can be constrained using the response of the system to a different forcing **B**, the response to which is potentially observable. Unlike constraints of the second kind, this logic does not require a specific feedback mechanism. Unlike constraints of the first kind (a special case), it is also not *a priori* true that the response of the system to one forcing **B** is controlled by the same processes which control the future response **A**. There are thus a larger number of potential sources of structural error compared to the first kind of constraint, as the simulation of responses to *both* **A** and **B** may have ensemble-wide biases and missing components. In this case, those potential biases may arise only in the simulation of the predictor or only the predictand, and so errors have the potential to weaken the constraint.

In such cases, the forcing associated with **B** differs from **A** in terms of its timescale or mechanism. Examples of this third kind of constraint have taken **B** as the seasonal cycle (Covey et al., 2000; Zhai et al., 2015b), the inter-annual variability simulated by the models (Cox et al., 2018b) (though it is arguable whether such unforced variability is in-fact measurable (Rypdal et al., 2018)) or the response to paleoclimate forcings (Hargreaves et al., 2012; Hegerl et al., 2006; Royer et al., 2007; Schmidt et al., 2014) or volcanic events (Boer et al., 2007; Plazzotta et al., 2018; Wigley, 2005). Similar approaches have used the seasonal cycle in snow albedo to constrain sea ice trends (Qu and Hall, 2014), future extreme precipitation (O'Gorman, 2012) and vegetation carbon responses to warming (Cox et al., 2013;





Wang et al., 2014; Wenzel et al., 2014).  The concept can be taken  further - using tendencies of forecasts on a timescale
of hours to constrain long term responses to climate change (Palmer, 2020; Rodwell and Palmer, 2007).

Because our confidence in the EC arises partly from the existence of the correlation within the ensemble itself, we
must carefully assess the possibility that the emergent relationship arises due to common assumptions which are
deployed throughout the ensemble.  Furthermore, it is more likely that a relationship will emerge if the common
assumptions are simple, with a small number of effective degrees of freedom in calibration (see Figure 2, in the simple-
model example which follows).

For example, many CMIP-class models use similar temperature-scaling assumptions for soil respiration (Shao et al.,
2013). There is evidence that the majority of soil carbon stocks in the CMIP5 archive can be explained by a reduced
order function of soil temperature and plant productivity (Todd-Brown et al., 2013), which notably fails to reproduce
observed carbon stocks - implying a common structural bias.  A constraint on the future temperature response in CMIP
(Cox et al., 2013) could be argued to effectively be a calibration of a low-order soil respiration model.

In such a situation, where the CMIP models have a common and/or low-order structure, differing only in their
calibration - the MME is in fact a PPE in disguise.  Our assumption that the ensemble represents a complete set of
plausible structural variants interchangeable with reality is far from the truth, and worse, an ensemble with such
structural limitations is more likely to produce constraints of the third kind (as we see in the simple example which
follows) because the response to any forcing is effectively governed by a small number of degrees of freedom.
Although there may be a robust intra-ensemble relationship between the response to a short-timescale forcing and a
long-timescale forcing, this relationship may be the direct product of a simple common structural framework.  In order
to have confidence in the constrained projection, it is then necessary to assess whether that common framework is
both adequate and also the only plausible mechanistic model for the process.

It should also be noted that these 'kinds' of constraint might be potentially useful in an illustrative sense, but they are
not absolute.  Some published constraints undoubtedly have elements of more than one type.  For example, (Zhai et
al., 2015b) has elements of both 2nd- and 3rd-kind constraints, in that it isolates a primary long term feedback process
and constrains it using the response to short term forcing (seasonal variability, in this case).  Another example is
constraining transient climate response from observed warming (Knutti and Tomassini, 2008; Nijsse et al., n.d.;
Schurer et al., 2018), which has elements of 1st- and 3rd-kind constraints. The transient warming response to an
idealized forcing is constrained with its response to historical emissions, which is a 1st-kind constraint But there are
also conceptual differences between these forcing pathways (most notably the presence of transient aerosol forcing in
the real world) and the resulting dominant feedback processes, which introduce elements of a third-kind constraint.
Ultimately, the greater the differences between the forced response considered in the constraint and that measured in
the predictand, the more the constraint itself depends on the structural assumptions present in the ensemble.

# 4 A SIMPLE EXAMPLE

We can illustrate these concepts using ensembles created from two different classes of simple climate model.

## 4.1 SIMPLE MODEL STRUCTURES

### 4.1.1 SINGLE-LAYER MODEL

The first model uses a single timescale of response, corresponding conceptually to an ocean represented by a single
thermodynamic slab:

$$C \frac{dT'}{dt} = F(t) - \lambda T',$$





where C is the heat capacity of the Earth system, T' is the global mean temperature anomaly, F is the time-dependent climate forcing and $\lambda$ is the climate sensitivity parameter.

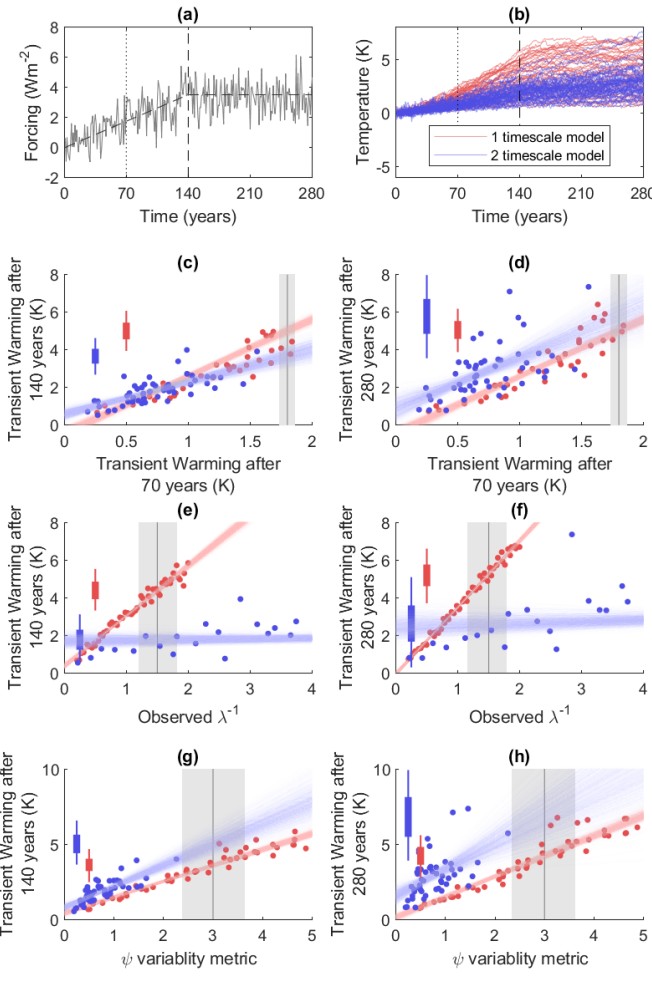


**Figure 2. An illustration of the 3 kinds of emergent constraint in two structurally-different ensembles.** (a) an idealised forcing timeseries used for each of the simulations - a (noisy) linear ramping of radiative forcing from years 0-140 followed by (noisy) constant forcing from years 140-280. (b) shows the response in 50-member perturbed parameter ensembles of two energy balance models, with 1 (red) and 2 (blue) timescales of response.

(c) a constraint of the first kind, showing TCR (warming after 70 years of 1 percent annual increase in CO2 concentrations) as a predictor of T140 (warming at time of CO2 quadrupling, 140 years in the same experiment). (d) warming after a further 140 years of constant (quadrupled) CO2 concentrations. (e,f) constraints of the second kind, using the feedback parameter 'lambda' to predict warming after (140, 280) years. (g,h) constraints of the third kind, using a variability metric (Cox et al., 2018b) derived from detrended temperature timeseries in years

1-70 as a predictor warming after (140, 280) years. In each case, colored points show members of the model ensemble, lines show bootstrap regression estimates, grey vertical bars show the 10th, 50th and 90th percentile of the (hypothetical) observed uncertainty distribution. Colored box/whisker plots show the 5/95th and 25/75th percentiles of the resulting constraint from each ensemble.



### 4.1.2 TWO-LAYER MODEL

The second model is slightly more complex, with the addition of a deep ocean (Geoffroy et al., 2013b):

$$C\frac{dT'}{dt} = F(t) - \lambda T' - \varepsilon\gamma(T' - T_0')$$

$$C_0\frac{dT_0'}{dt} = \gamma(T' - T_0'),$$

Where $C_0$ is the heat capacity and $T_0'$ is the temperature anomaly of a deep ocean layer, $\gamma$ is the thermal diffusion coefficient of heat exchange between the two layers, and $\varepsilon$ is the efficacy of heat transfer to the deep ocean (see
(Geoffroy et al., 2013b)).

### 4.2 IDEALIZED EXPERIMENTS

We conduct an idealised climate change experiment where for the first 140 years, $CO_2$ concentrations are increased by 1 percent each year resulting in a gradual linear increase in forcing over time, followed by an equilibration period:

$$F(t) = at + b\eta(t), t < 140.$$

A transient component of the forcing is provided by the first term, where a=0.05 (corresponding approximately to the 1 percent $CO_2$ ramping experiment), and a random component is provided by the second term, where $\eta(t)$ is white Gaussian noise, scaled by the factor b=0.5. In the second 140 years, the transient component of the forcing is held constant:

$$F(t) = 140\,a + b\eta(t), t >= 140,$$

With each model, we produce a range of responses by creating an ensemble with parameters sampled in latin hypercube - in the first case $[C, \lambda]$ and in the second case, $[C, C_o, \lambda, \varepsilon, \gamma]$. Finally, we consider how different types of artificial 'observation' would constrain the projected response. Parameter ranges for the two-layer model are informed by (Geoffroy et al., 2013b), and manually adjusted in the one-layer model to produce a comparable range of transient warming after 140 years (T140 herein, see Table 2).

| Parameter | Symbol (Units) | Minimum (1 layer model) | Maximum (1 layer model) | Minimum (2 layer model) | Maximum (2 layer model) |
|---|---|---|---|---|---|
| Upper ocean heat capacity | C (Wm⁻²K⁻¹yr) | 10 | 20 | 2 | 10 |
| Feedback parameter | $\lambda$ (Wm⁻²K⁻¹) | 0.5 | 2 | 0.5 | 5 |
| Deep ocean heat capacity | C₀ (Wm⁻²K⁻¹yr) | - | - | 50 | 500 |
| Deep ocean diffusion coefficient | $\gamma$(Wm⁻²K⁻¹) | - | - | 0.5 | 3 |
| Deep ocean efficacy | $\varepsilon$(unitless) | - | - | 0.8 | 2.5 |

**Table 2. Parameters used in the one- and two-layer models in the idealized example, and the upper and lower bounds of the sampling range used in the ensemble construction.**



In these simple models, we can test constraints of different types and illustrate their sensitivity to common structural differences between the two ensembles. We consider three constraints for future response in each of these models, and then interpret their relative skill.

A 1st-kind constraint can be created by using the transient warming observed after 70 years (T70) to predict T140. In this example, the EC exists in both ensembles (though its slope differs a little between the two ensemble types): transient warming is near-linear in time in both cases, and so behaviour at year 140 can be extrapolated from years 1-70. However, for the case of warming at 280 years (T280, i.e. an additional 140 years after forcing is stabilized), we see a meaningful constraint only in the single layer model (Figure 2b). In the two layer model, the temperature
response in the first 140 years of linear forcing increase is a combination of both slow (deep) and fast (shallow) timescale components, and transient warming at year 70 can be extrapolated (even if we don't know the relative contribution of the slow and fast components of the warming). However, when the forcing stabilizes at year 140, the shallow component quickly saturates and remaining warming is due to deep ocean equilibration alone. Thus, this additional degree of freedom (shallow vs deep contribution to transient warming) is unconstrained, and T70 is a worse
constraint on T280. The one layer model does not have this additional degree of freedom, and thus T70 is a good constraint on T280 - but only because of the structural simplifications present in the model. Because the nature of the forcing differs between the transient and equilibrium stages of the experiment, the constraint of T280 using T70 is a 3rd-kind constraint in our classification system.

We can consider a constraint of the 2nd kind by assessing how independent data constraining a parameter in the
models would constrain its projections. In Figures 2(e,f) we illustrate how knowledge of the $\lambda$ parameter would act as a constraint in two ensembles (as a proxy for information about physical processes in CMIP-class models). In the single-timescale model, $\lambda$ acts as a near-perfect predictor of warming after 140 and 280 years, and constraining ensemble spread using that parameter would have a large effect. In contrast, in the two-timescale model, the correlation is weak. Although the lambda parameter controls feedbacks (and equilibrium response) in both models,
transient response in the two layer model is strongly governed by deep ocean heat uptake. We know that heat uptake by the deep ocean is an important mechanism for Earth's warming in transient scenarios (Geoffroy et al., 2013a), so we have introduced a common structural flaw in models that do not account for the role of the deep ocean. That flaw allows for an apparently strong EC in the single-timescale model ensemble which is not present in the more realistic ensemble.

The one-layer model ensemble samples a similar range of transient warming as the two-layer model in the first 140 years. For some applications, the one-layer model may be sufficient to model further transient warming, but the strength of an EC based on $\lambda$ depends on the over-simplistic structure of the one layer model, which leads to a demonstrably overconfident result in this case.

We can also construct a 3rd kind constraint such as the $\psi$ variability metric similar to that used by (Cox et al., 2018b),
where the variance and lag-covariance of temperature variability is used as a predictor of climate sensitivity (though there are conceptual differences to Cox 2018, given our model here does not have an internal source of noise). In this case In Figure 2(g,h), we consider the $\psi$ metric as a predictor of T140 and T280 in our two ensembles. Once again, the metric is a strong predictor for both T140 and T280 in the one-layer ensemble. Meanwhile, in the two layer ensemble, the correlation with T140 is weaker (with a different slope to the one-layer case). There is little to no
correlation between T280 and $\psi$. As with the first-kind constraint, both the EC relationship slope and its strength as a predictor depend on common structural assumptions, with a stronger apparent relationship in the ensemble with fewer degrees of freedom.

In these simple examples, we can understand EC behavior in the context of the model assumptions. Both model types can produce similar transient evolution until forcing is fixed, but then the responses diverge, revealing very different
equilibration behaviour (see Figure 2b). The single-layer model equilibrates to a change in forcing over 1-2 decades (depending on the exact choices of C and $\lambda$), so that after 140 years, most of the response to the forcing experienced



to date has already been realized in the model temperature response, and little additional warming is subsequently seen. T70, T140 and T280 are all (to first order) controlled by the $\lambda$ parameter. On the other hand, the-two layer model does not fully equilibrate to a step change in forcing for centuries - so the transient response to forcing which define T70 and T140 is controlled by both $\lambda$ and the deep ocean heat uptake parameters ($C_o, \varepsilon, \gamma$). In this model, neither T70 nor $\lambda$ are singularly informative about how the model equilibrates.

This illustrates a key issue with the emergent constraint framework: if one has access only to the one-layer model ensemble, one would conclude that $\lambda$ or T70 are strong emergent constraints on T280, and the strength of the relationship might be used as evidence for the physical plausibility of the EC. But instead, in this case, the strength of the relationship is indicative that the single layer model is lacking (in this case a deep ocean), and the parameters of the shallow ocean have been adjusted to compensate for this bias in reproducing observed transient behavior. Furthermore, if independent data on the real-world value of $\lambda$ was available and was used to constrain the response of the single-layer model (and the real world was in fact more appropriately modeled by including the deep ocean), the resulting constrained prediction would be precise but inaccurate because that prediction would be conditional on a common structural assumption that is incorrect.

## 5 ASSESSING STRUCTURAL ROBUSTNESS IN CMIP EMERGENT CONSTRAINTS

Clearly, the models in the example above are vastly simpler than those used in CMIP, but these examples illustrate relationships which could emerge in those more complex models, and how they might be incorrectly utilized. Such errors could occur in CMIP-derived ECs if there are processes that are parameterised in a common, overly-simplistic fashion across the ensemble. Furthermore, irrespective of increasing model complexity, it is likely that this argument could always be made - one could *always* imagine a more complex or complete model than the standard at any given time (e.g., turbulence closure). Similarly, the reverse could occur - the presence of models which oversimplify or erroneously represent a process could mask a constraint which might exist in the remaining ensemble. In this context, a single EC will continue to be at best a conditional statement which could be proved inaccurate or overconfident by the following generation of models.

But for the increasing body of ECs which have been published using CMIP data, how concerned should we be about overconfidence due to common structural errors? This question does not replace those credibility tests which have already been proposed in the literature (Caldwell et al., 2018; Hall et al., 2019): robustness to change in ensemble samples, plausibility of mechanism and evidence of the mechanism and feedback variability from supporting model diagnostics. But for ECs which appear to pass these tests, an assessment of the underlying model assumptions is then necessary. Here we assess a small number of ECs as case studies, and how their applicability is to some degree conditional on structural assumptions in the source ensemble.

### 5.1 PERSISTENT BIAS OF CO2 CONCENTRATIONS

We consider first an example of an EC of the 'first kind' (Hoffman et al., 2014) which uses the present day carbon dioxide concentration to constrain future carbon dioxide concentrations. Their primary finding is that a historical bias persists into the future in a transient emissions scenario. Given variability in transient carbon dioxide concentrations (both in the present and future) in CMIP5 is primarily a function of land surface carbon-concentration and carbon-climate feedbacks (Friedlingstein et al., 2014), this exploitation of bias-persistence might be be overconfident if the CMIP5 models were missing or misrepresenting key land surface processes which might differently alter future and historical CO2 concentrations.

There are a number of such processes missing from a subset or the entirety of the CMIP5 ensemble. For example, nitrogen limitation was implemented in only one model in the CMIP5 generation of models (Zaehle et al., 2015), where it was found to have the capacity to significantly alter land carbon uptake. For an emergent constraint exploiting





the persistence of a $CO_2$ concentration bias, this is potentially an issue if nitrogen availability is not currently limiting, but becomes a limiting factor in a future state. A larger fraction of CMIP6 models include nitrogen limitation with diverse implementations. Nitrogen was not found to strongly influence historical carbon uptake - but a future effect has not been explicitly ruled out by studies to date (Davies-Barnard et al., 2020), so a repeat of the Hoffman study would be a useful test of the robustness of the EC to a significant structural change between the CMIP5 and CMIP6

generation of land surface models.

There remain a number of additional processes which could potentially influence future carbon uptake that are not comprehensively implemented. Phosphorus limitation has potential large impacts on the future Amazonian carbon sink (Fleischer et al., 2019), and is absent from CMIP5 models, but present in a small subset of CMIP6 models (Arora et al., 2020). The impact on the carbon sink of potential changes in tree mortality in response to CO2 and forest

productivity. is both critical and absent from CMIP6 class models (Brienen et al., 2020; Needham et al., 2020), as are complex fire-vegetation feedback processes (Teckentrup et al., 2019), diversity in responses to drought (Fisher et al., 2010; Levine et al., 2016; Longo et al., 2018; Sakschewski et al., 2016) vegetation damage under unprecedented heat extremes (Teskey et al., 2015), wind events and pathogen damage (McDowell et al., 2018). These all have the potential to introduce climate-vegetation feedbacks which are currently not represented in the CMIP6 ensemble.

Thus our confidence in the persistence of the models' present day $CO_2$ bias persisting into the future is reduced because there are processes which are potentially highly significant and are broadly absent from current generation models. However, the nature of a first-kind constraint means that the integrative carbon cycle response is used as both predictor and predictand, and so this kind of constraint could remain robust as long as structural omissions had similar effects on $CO_2$ concentrations in the past and the future. In short, it is a filter on models which have been accurate

thus far in simulating the quantity we are ultimately interested in measuring - an arguably necessary (but not sufficient) condition for projecting that quantity into the future. Because the net carbon feedback is being constrained directly, the method is (somewhat) insensitive to the representation of processes which make up that feedback.

## 5.2 HISTORICAL CONSTRAINTS ON SOIL-CARBON TEMPERATURE RELATIONSHIPS

We next consider Cox et al. (2013), which relates tropical land carbon uptake-temperature feedback and the historical

relationship between growth rate of atmospheric $CO_2$ and tropical temperature anomalies (other studies (Chadburn et al., 2017; Varney et al., 2020) have considered similar relationships using spatial variability as a predictor). In CMIP5 models, this constraint (of the 'third-kind' ) was very strong (Cox et al., 2013). In this case, the focus on the carbon-temperature component of the total carbon feedback isolated the effect of soil respiration temperature response - which in CMIP5 dominates both the predictor and the predictand for the EC. Our confidence in the EC thus firstly depends

on whether soil respiration is represented in a common, oversimplified fashion in the CMIP5 ensemble. Independent studies have found that inter-model differences in soil carbon uptake are dominated by the parameterisation choices for soil heterotrophic respiration rather than structural differences (Todd-Brown et al., 2013), and that a lack of ability to represent grid-scale variation in soil carbon levels indicates the potential missing processes. Non-coupled models representing higher levels of microbial complexity and vertical resolution suggest that CMIP-class models may be

underestimating the range of potential future soil carbon uptake (Shi et al., 2018).

In CMIP6 models, there remains indication that spatial variability continues to provide predictive information on future soil carbon dynamics (Varney et al., 2020), but the role of soil respiration in the total carbon-temperature feedback is less dominant (Arora et al., 2020), with vegetation productivity responses also playing a role in the ensemble variance. This increases the structural diversity of the relevant model components, and has the potential to

weaken the strength of the CMIP5 correlation. A repeat of (Cox et al., 2013) for the CMIP6 ensemble would be therefore of interest for testing whether the correlation remains equally strong in CMIP6.

Unlike Hoffman et al., (2014), Cox et al. (2013) is a 3rd-kind constraint. The predictor is a function of different spatial and temporal forcing scales to the predictand, and the real world may contain additional unresolved processes which




influence one, but not the other. Unlike a first-kind constraint, the net long term climate change carbon feedback is
not being constrained directly, it is constrained by proxy. The addition of additional components representing
currently unresolved land surface processes in future ensembles could therefore have the potential to change the value
of the predictor and predictand independently, which could bias constrained carbon cycle feedback estimates derived
from the application of the constraint to observations in the current ensemble and also potentially weaken the strength
of the relationship itself in future ensembles.

## 5.3 Constraining transient climate response with observed warming

The constraint of TCR detailed in Nijsse et al. (2020) (and also Tokarska et al., 2020) use observed transient warming
as a predictor of future warming. In this case, the EC falls into the 'first kind' category - the predictor and predictand
are conceptually similar in that they both represent the transient global mean warming response to a CO2 forcing
which is monotonically increasing at broadly comparable rates - but there are differences in terms of the forcing
magnitude (present day $CO_2$ levels are less than the double pre-industrial level used in the formal TCR definition),
and also due to other forcing terms due to, for example, aerosols and land use change. The authors minimise the role
of aerosol forcing changes by considering a time period (1975 to 2013) in which there is relatively constant global
mean aerosol forcing - leaving a time period in which greenhouse gas forcing changes are dominant.

The presence of a strong correlation in CMIP6 indicates that, at least in this ensemble, transient warming remains
broadly constant in response to linearly increasing forcing, and uncertainties in the extrapolation of transient warming
are sufficiently small that the inter-model spread of TCR can be constrained. Unlike equilibrium response (where
models show rather diverse equilibrium warming trajectories (Rugenstein et al., 2020)), CMIP models tend to
uniformly exhibit near-linear warming trajectories in response to transient forcing, differing only in the temperature
growth rate - thus making a strong constraint with effectively one degree of freedom.

As such, the constraint of TCR from observed warming in a period where primarily only greenhouse gas forcing is
changing is likely to be quite robust, leaving the primary question of the utility of TCR itself as a metric of response
in future projections. The TCR metric is insensitive to carbon cycle dynamics and aerosol forcing plus potential
'tipping points' (Lenton et al., 2019) if they are unrepresented in current generation models. TCR is also a combined
function of climate feedbacks and ocean heat uptake dynamics, and models which share the same value of TCR can
have different warming trajectories long after forcing levels stabilise (Sanderson, 2020). As such, it seems that an
emergent constraint on TCR and warming until 2100 in realistic scenarios might be robust, but may not constrain post-
2100 warming under mitigation.

## 5.4 Process-based constraints on climate sensitivity

Here, we consider an example of a 2nd kind process constraint (Sherwood et al., 2014) on equilibrium climate
sensitivity in CMIP5 - though the arguments would be equally applicable to other plausible process-based constraints
(Brient et al., 2016; Brient and Schneider, 2016; Zhai et al., 2015a). The Sherwood paper proposes two indirect
metrics of lower tropospheric mixing which are related to future reductions in boundary layer clouds (the cloud
feedback which is itself the largest component of inter-model spread in ECS (Pincus et al., 2018)). The postulated
physical mechanism is that models with larger boundary layer mixing will experience stronger ventilation of moisture
from the lower troposphere as the atmosphere warms and humidity increases, so these models ultimately experience
the most extreme loss of boundary layer clouds. Independent studies have assessed the Sherwood constraints to have
a plausible mechanism, with correlated warming patterns occurring in regions which are consistent with the constraint
(Brient, 2019; Caldwell et al., 2018). Together with the relatively strong correlation which is seen in the Sherwood
paper itself, this makes the study one of the more compelling examples of a physical constraint on ECS in a multi-
model ensemble.




If indeed the Sherwood constraint is a robust predictor of ECS within CMIP5, the structural robustness of the constraint concerns the degree to which CMIP5 is a representative sample for comparison with reality. This question can itself be divided into three questions: (1) is the process itself sufficiently well represented in CMIP5 to be informative, (2) are there other processes which are absent, undersampled or commonly misrepresented in CMIP5 models which might bias ECS and (3) are there common structural biases which might impact the predictors - the mixing proxies in this case, thus biasing the conclusion of the constraint.

For the first question of boundary layer process accuracy, there is a structurally rich selection of boundary layer schemes in CMIP5 (Edwards et al., 2020) which reduces the chance that the EC is a product of structural homogeneity in the ensemble. There is, however, some evidence that there exist ensemble-wide climatological biases in current generation models which can be attributed to common boundary layer mixing structural errors in CMIP5 (Wei et al., 2017) and most CMIP5 generation models rely on low-order turbulence closure schemes which assume, to some degree, a representative length scale for temperature and wind gradients based on Monin-Obukhov similarity theory (Monin and Obukhov, 1957) (often complemented by bulk convection schemes or energy closure arguments to resolve remaining boundary layer mixing). The testing of the persistence of the EC in CMIP6, which includes models with higher order closure schemes which do not make this explicit assumption (Bogenschutz et al., n.d.), thus broadening the diversity of representation of boundary layer mixing in the ensemble and creating a useful test of structural robustness for the CMIP5 era constraints.

The second question relates less to the representation of the process in question (shallow convection and boundary layer processes), and more to everything else in the model which could potentially influence ECS in CMIP5, but might be undersampled (or not represented at all). To put this another way, are boundary layer processes responsible for ECS variation in CMIP5 because they are the most uncertain in an absolute sense, or because we have failed to adequately explore uncertainty in other feedback processes? For example, the transition from CMIP5 to CMIP6 saw many models shift in their representation of mixed-phase clouds which are thought to explain high ECS values in a number of CMIP6 models (Zelinka et al., 2020), so it is unclear whether the Sherwood constraint would represent that shift given the process responsible differs from the primary axis of CMIP5 variability.

Perturbed parameter experiments have reported ranges in ECS which have been dominated by deep convective (Sanderson et al., 2010) or mid-layer cloud response (Shiogama et al., 2012), and hence it is not surprising that the Sherwood constraint on low cloud feedbacks has proven less effective at constraining ECS in a PPE (Kamae et al., 2016). If the range of deep convective and mid-layer cloud feedbacks seen in these PPEs cannot be otherwise ruled out, this raises a concern for the degree to which CMIP5 has sampled the climate feedback space, and thus structural robustness of the Sherwood 2014 constraint used in isolation.

The final question for process-based constraints is the degree to which predictive metrics in the ensemble could be biased by the omission or misrepresentation of other processes. For boundary layer measurements in CMIP5, biases in the land surface scheme are known to project onto boundary layer climatologies (Holtslag et al., 2007), which in the case of CMIP5 was responsible for ensemble-wide systematic biases due to common soil moisture biases (Svensson and Lindvall, 2015) - but given that the Sherwood constraint is focussed on ocean, it seems unlikely that these effects are highly influential. However, biases in boundary layer simulation have been attributed to cloud morphology (Bony et al., 2020), large scale flow, gravity wave and surface drag parameterizations (Sandu et al., 2013), so there remains the possibility of an ensemble-wide bias in the predictor if any of these processes are commonly misrepresented.

## 5.5 CONSTRAINING CLIMATE SENSITIVITY WITH FLUCTUATION-DISSIPATION RELATIONSHIPS

We finally consider a second-kind constraint on ECS (Cox et al., 2018b) which relates a metric of internal variability (Psi, a function of the lag-covariance structure of the global mean temperature timeseries) to the models' ECS. The





constraint exploits the fluctuation-dissipation theorem (Kubo, 1966; Leith, 1975), which relates the linear response of a dynamical system to its noise characteristics. The result is somewhat dependent on subjective choices in the derivation of the unforced lag-covariance term (Brown et al., 2018), the length of sample used (Rypdal et al., 2018), the subset of CMIP5 models used in the ensemble (Po-Chedley et al., 2018) - which together might imply that there
are uncertainties involved in the practical application of the constraint using the historical record which were not represented in the original study.

Setting aside for a moment these practical issues associated with measuring unforced variability in reality - there is reasonable evidence that there might exist a relationship between control model variability and climate sensitivity in the CMIP5 ensemble (Cox et al., 2018a) (whether that unforced variability is measurable in practise is a different
question). The fact that this idealised relationship exists both in simple models (Williamson et al., 2019), and in the CMIP5 ensemble (where both internal variability and ECS are emergent properties of a large number of interacting processes which are diversely sampled within the ensemble) makes it reasonably unlikely that the EC is conditional on strong common structural assumptions. A test of the relationship between control variability and ECS in CMIP6 would nevertheless help confirm this hypothesis.

Understanding the disagreement between a number of plausible (Caldwell et al., 2018) process-based ECs which constrain ECS to higher values (Brient and Schneider, 2016; Sherwood et al., 2014; Zhai et al., 2015b) and fluctuation-dissipation arguments which suggest lower values (Cox et al., 2018b) may thus require a joint consideration of structural and implementation errors. The process constraints are strongly conditional on the sampling of feedback processes in the CMIP ensemble itself. If the CMIP5 ensemble is under-sampling other types of radiative feedback
(e.g. deep convection, mid-level cloud response), then this uncertainty is not represented within the constrained distribution obtained from using an EC on boundary layer processes. Such structural uncertainty is less applicable to the fluctuation-dissipation constraint because the variability of global mean temperature is an integrative property of all feedbacks in the system, it is less conditional on any single feedback type being well sampled in the ensemble.

However, the practical limitations of the short historical record confounded by other climate forcers may prevent its
useful application in practise because the unforced variability of the system is not sufficiently knowable to form a strong constraint on ECS. The results may also be sensitive to the metric and the set of models used; an earlier study using a similar idea found no constraint (Masson and Knutti, 2013), and in some cases reversed signs of correlations between CMIP and PPEs, thus questioning the robustness of the approach. Other studies (Annan et al., 2020) have performed objective Bayesian constraint of ECS through climate variability in simple models, finding a wider
constrained range wider than suggested by Cox 2018. As such, a confirmation of the strength of the Cox 2018 relationship under CMIP6 would provide valuable additional data on its robustness.

## 6 CONCLUSIONS

We have highlighted here that the existence of disagreement among published constraints suggests that structural errors exist in the CMIP multi-model ensemble, and that some published constraints may be spurious. A common
structure in the ensemble may lead to strong EC relationships, especially if assumptions have only a small number of degrees of freedom - and that such situations may indicate a lack of structural diversity which might be necessary for robust uncertainty quantification.

It remains to consider how an assessment of potential structural errors in an emergent constraint should be used. The focus of published papers and their use in, e.g., IPCC assessments, has often been on the constrained result itself (Cox
et al., 2013, 2018b), but these constraints may be overconfident in the face of a potential or demonstrated structural error. A more robust interpretation of an EC is that it provides potentially observable information related to aspects of ensemble response variation, but not necessarily that the projection can be accurately constrained directly with that information. In our simple example, given the presence of a relationship between $\lambda$ and T280 in the single-layer





ensemble, it might be accurate to interpret that the processes represented within $\lambda$ could be relevant to long term temperature evolution, but unjustified to actually constrain T280 directly.

If this logic is applied to the more complex models which are used in climate assessments, such information could potentially highlight which processes control ensemble spread in projections, where model development needs to assess whether current process representations are adequate and appropriately diverse, whether there are alternative process models which could be incorporated into CMIP-class models, and where available observations have not been fully exploited to calibrate models.

This information could also motivate more focus on the simulation of the predictor variable - are there processes which are missing in the current generation of models which could be implemented in future versions? The presence of an emergent constraint should also act as a warning sign that a process in the ensemble may be represented in a structurally homogeneous fashion. Such an effect could be compounded if there are only a small number of effective degrees of freedom sampled in the ensemble. It is thus critical to assess whether common simplifications in the ensemble are creating or influencing emergent relationships.

In any regression, the points to the extreme end of the predictor distribution have greater 'leverage' on the estimation of the regression relationship (Chatterjee and Hadi, 1986), which means that the models with the most influence in defining an EC are potentially those with the greatest errors in the relevant metric. Similarly, some ECs might be 'hidden' because outliers weaken a relationship which may exist in a subset of models. In either case, correct interpretation of the EC would require an independent assessment of the plausibility of the participating ensemble members and the degree to which the ensemble samples potential degrees of freedom in the modeling of relevant processes - which requires the consideration of other metrics in addition to the EC itself.

The use of an EC as the sole constraint of a projected quantity is effectively a model weighting which considers only a subset of model performance, disregarding aspects of model performance which are not represented within the EC itself (even though that one metric may characterize many aspects of the climate, or itself be a sum of different metrics). This should give us pause, because studies of model weighting have demonstrated that using a single metric that only captures specific aspects of climate is likely to result in an overconfident result (Knutti et al., 2017; Lorenz et al., 2018). As such, care must be taken to recognise that even if an EC exists, structural biases may preclude a simple assessment that those models closest to the observed value have the most trustworthy response. For example, if calibration trade-offs prevent models from being tuned to match observations in two locations simultaneously, this may complicate the application of an emergent constraint which uses simulated climate in one of those locations as a predictor of response.

The development of multi-metric approaches (Bretherton and Caldwell, 2020; Brient, 2019; Brunner et al., 2020; Huber et al., 2011) could provide greater robustness to structural errors, given that a lesser reliance is placed on any single axis of inter-model variability. Even if two constraints are identified for the same physical process, and the metrics are highly correlated within the ensemble (Caldwell et al., 2018), there may be some advantage in combining their results, given the potential for differing and potentially independent biases in observations of the two quantities (Lorenz et al., 2018). Though uncertainty in observational products themselves must still be sampled where possible, multi-metric approaches have the potential to reduce observational uncertainty on constraints (Brunner et al., 2019).

The idea of multi-variate metrics of model performance is not new, and generic multi-variate metrics of model climatological errors are perhaps the default approach for assessing the skill and plausibility of different models during assessment (Baker and Taylor, 2016; Gleckler et al., 2008; Wilde et al., 2011). But, weighting models based on general climatological performance over a large number of variables has little effect (Sanderson et al., 2017) which does not tend to significantly decrease the projection uncertainty in the unweighted ensemble.


ECs could play a useful role by defining reduced-space metrics which consider only those aspects of model performance that are relevant to a particular future response. Multi-metric emergent constraints may provide a useful 'third way': they are less sensitive to structural errors than single-metric emergent constraints, and can be targeted toward processes that may drive future responses more accurately than generic performance metrics which do not explicitly account for the relevance of an observable to a given response (Baker and Taylor, 2016; Collier et al., 2018).

There is undoubtedly also rich information to be gained from ECs which disagree - a rare quantitative indicator of projection-relevant structural error in climate model simulations. If inconsistent constraints are proven to be statistically robust, these inconsistencies could provide guidance in future development cycles - highlighting key biases shared among models related to missing or misrepresented processes which might be important in properly representing feedbacks of interest.

The collection of simulations and projections available in CMIP represents a formidable amount of data (Williams et al., 2016), but its scale does not justify considering CMIP to be a comprehensive sample of possible representations of the Earth System. Parametric uncertainties and computational limitations on resolution and ensemble size limit the degree to which our current ensembles represent the tails of the distribution of possible future change, and any statement of uncertainty of the future evolution of the climate system can only be made robustly in the context of these uncertainties. Emergent constraints, if used less literally, could play a powerful role in understanding the ensemble we have; a combination of more robust statistical frameworks, better understanding of the ensemble's nature and multi-metric techniques could provide new opportunities for understanding how the Earth might respond to climate forcing.

## CODE AVAILABILITY

Code for this study is available on an open github repository at https://github.com/benmsanderson/structure_ec

## DATA AVAILABILITY

Data for Figure 1 uses published values directly from the citations listed in Table 1.

## AUTHOR CONTRIBUTION

Benjamin Sanderson performed analyses and wrote the main text. Angeline Pendergrass provided text feedback and suggestions for analysis. Ben Booth provided text feedback and helped develop the conceptual basis for paper. Charles Koven, Florent Brient, Rosie Fisher provided text feedback. Reto Knutti provided extensive text feedback.

## COMPETING INTERESTS

The authors declare no competing interests

## ACKNOWLEDGEMENTS

Thanks to Laurent Terray for useful conversations in developing this study. Dr Sanderson was funded by the French National Research Agency, grant number ANR-17-MPGA-0016. This material is based in part upon work supported by the National Center for Atmospheric Research, which is a major facility sponsored by the National Science Foundation (NSF) under Cooperative Agreement 1947282. Portions of this study were supported by the Regional and Global Model Analysis (RGMA) component of the Earth and Environmental System Modeling Program of the U.S. Department of Energy's Office of Biological & Environmental Research (BER) via NSF IA 1844590.



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






| Model | Sensitivity | Qu 2013 | klein 2013 | Su RH | Su alpha | Sherwood S | Sherwood D | Briant Shal | Zhai | Tian | Brient albedo | Lipat | Cox |
|---|---|---|---|---|---|---|---|---|---|---|---|---|---|
| *Observed max* | | *-0.019* | | *1.25* | *1.25* | *0.48* | *0.44* | *49* | *-0.72* | *1.5\** | *-0.74* | *-34.83\** | *0.145* |
| *Observed min* | | *-0.034* | | *0.75* | *0.75* | *0.35* | *0.38* | *41* | *-1.85* | *0.5\** | *-1.18* | *-36.83\** | *0.113* |
| ACCESS1.0 | 3.8 | | | | | 0.4 | 0.31 | | | 0.77 | -1.37 | | 0.22 |
| ACCESS1.3 | 3.5 | | | | | 0.36 | 0.41 | | | 0.98 | -0.97 | | |
| UKMO-HadCM3 | 3.3 | -0.001 | 1.51 | | | 0.41 | 0.2 | | -0.88 | 0.53 | | | |
| UKMO-HadGEM1 | 4.4 | | | | | 0.39 | 0.35 | | | 0.49 | | | |
| BCC-CSM1.1 | 2.9 | | | | | 0.42 | 0.21 | 47 | | 1.42 | -0.24 | -33.01 | 0.19 |
| BCC-CSM1.1m | 2.9 | | | | | | | 49 | | | -0.58 | -35.01 | 0.14 |
| BNU-ESM | 4.1 | | | | | 0.41 | 0.19 | 48 | | 1.54 | -0.72 | | 0.15 |
| CESM-CAM5 | 4.1 | | | 0.92 | 0.8 | 0.41 | 0.31 | | -1.1 | | -0.3 | | |
| CCSM4 | 2.9 | 0.003 | 1.27 | | | 0.38 | 0.24 | 49 | | 1.27 | -0.29 | -36.71 | 0.19 |
| CNRM-CM5 | 3.25 | | 1.26 | | | 0.33 | 0.25 | | -0.38 | 1.56 | -0.23 | -33.55 | 0.16 |
| CSIRO-Mk3.6 | 4.08 | -0.0002 | | 0.81 | 1.2 | 0.36 | 0.43 | | -0.77 | 0.97 | -1.17 | -34.26 | 0.21 |
| CanESM2 | 3.69 | -0.002 | 1.027 | 0.7 | 0.76 | 0.31 | 0.21 | 74 | -1.09 | 0.9 | -0.74 | -33.25 | 0.17 |
| FGOALS-g2 | 3.5 | | | | | 0.51 | 0.17 | | -0.44 | 1.14 | -0.27 | | |
| FGOALS-s2 | 4.2 | -0.005 | | | | 0.59 | 0.16 | | | | -1.13 | -30.74 | |
| GFDL-CM3 | 3.97 | 0.004 | 0.91 | 0.87 | 1.23 | 0.36 | 0.24 | 44 | -2.09 | 1.48 | -0.8 | -34.09 | 0.36 |
| GFDL-ESM2G | 2.39 | -0.005 | | 0.37 | 0.56 | 0.45 | 0.17 | 38 | | 1.06 | -0.26 | -35.53 | 0.2 |
| GFDL-ESM2M | 2.4 | -0.003 | | | | 0.46 | 0.16 | 39 | | 1.52 | -0.35 | -34.77 | 0.15 |
| GISS-E2-H | 2.3 | | | | | 0.3 | 0.22 | 18 | -0.17 | 2.54 | 1.14 | | 0.1 |
| GISS-E2-R | 2.1 | -0.018 | | | | 0.28 | 0.23 | 19 | 0 | 2.05 | 0.84 | -32.98 | 0.12 |
| HadGEM2-CC | | -0.016 | 0.81 | 0.58 | 0.42 | | | | | | | | |
| HadGEM2-ES | 4.59 | -0.021 | | | | | | 4 | 4.6 | 1.04 | -1.7 | -33.63 | 0.29 |
| INM-CM4.0 | 2.1 | | | 0.34 | 0.56 | 0.33 | 0.13 | 89 | 0.75 | 2.04 | -0.13 | -35.31 | 0.07 |
| IPSL-CM5A-LR | 4.13 | -0.025 | | 0.7 | 0.95 | 0.45 | 0.27 | 70 | -1.26 | 1.01 | -0.91 | -30.41 | 0.2 |
| IPSL-CM5A-MR | 4.12 | | | | | | | | | | -1.07 | | |
| IPSL-CM5B-LR | 2.6 | | | | | 0.42 | 0.15 | 83 | | 1.33 | -1 | -30.73 | 0.16 |
| MIROC-ESM | 4.67 | -0.024 | | 0.92 | 0.94 | 0.56 | 0.22 | 74 | -1.16 | 0 | -0.78 | -31.48 | 0.23 |
| MIROC-ESMCHEM | | -0.033 | | | | | | | | | | | |
| MIROC5 | 2.72 | -0.013 | 1.47 | 0.7 | 0.45 | 0.45 | 0.26 | 38 | 0.22 | 0.66 | -0.33 | -34.68 | 0.23 |
| MPI-ESM-LR | 3.63 | -0.018 | 1.03 | 0.85 | 0.91 | 0.47 | 0.23 | 70 | -0.67 | 0.51 | -0.53 | -33.8 | 0.15 |
| MPI-ESM-MR | 3.4 | | | | | 0.46 | 0.23 | 72 | | 0.61 | -0.51 | | |
| MPI-ESM-P | 3.4 | | | | | 0.46 | 0.23 | | | 0.39 | -0.63 | -33.76 | |
| MRI-CGCM3 | 2.6 | 0.014 | 0.98 | 0.79 | 0.4 | 0.41 | 0.27 | 35 | 0.41 | 1.87 | -0.71 | -34.18 | 0.09 |
| NorESM1-ME | | | | | | 0.39 | 0.25 | | | | | -36.16 | |
| NorESM1-M | 2.8 | 0.002 | | 0.84 | 0.61 | 0.38 | 0.24 | 47 | 0.31 | 0.91 | -0.41 | -36.23 | 0.14 |

**Table 3.** *ECS values, and emergent constraint values used in Figure 1, for constraints described in Table 1. Observed values are as originally published, apart from starred values where uncertainties in observations were not published and estimated here as +/- 50 percent (Tian) or +/- 1 degree (Lipat).*