# Peer review of "On structural errors in emergent constraints"

_Earth System Dynamics, 2020_

## Author Response (AR1)

**On structural errors in emergent constraints**

Response to reviewer 1

May 25, 2021

*The authors have assessed the application of emergent constraints to estimate uncertainties in unknown climate projections. The recent increase in the application of emergent constraints in Earth Science makes this a timely issue, even more in the last year since the new ensemble of Earth system models (CMIP6) became available. Furthermore, the manuscript is very well written and structured, which made it a pleasure to read.*

Many thanks to the reviewer for the kind comments and thoughtful review.

*However, the main motivation for the paper is that emergent constraints are used too literally (line 681) and "confusing to policymakers" (line 17). I do not fully share this assumption and I argue below why I think this is not the case.*

We argue below why we find this position to be justified.

**1 Major comments:**

*(1) The manuscript reads more like a review and less like a research article. The analysis in this paper includes (a) calculating correlations between different published variables, (b) a bootstrapping approach to exploit previously published emergent constraints, and (c) the exploitation of two differential equations with a random number generator to create many ensemble members.*

*Taken together, I cannot see how this would be a sufficiently novel concept, idea, tool, or data given that was not previously exploited. Point (b) somehow tests the robustness of the linear relationship. What is the advantage of this method compared to the published measures of uncertainty (e.g. prediction intervals, see minor comment 1). Already the published results of all shown ECS are not different within their still relatively large uncertainties (Table 4*

*in Schlund et al. (2020)). So, I am wondering what additional information is obtained by performing this bootstrapping. Point (c) seems to be a more of a fancy way of saying that the Earth System responds at different timescales and not just one, a well-known concept and used to calculate the temperature response to radiative forcing changes (Stocker et al., 2013; Otto et al., 2015). It is therefore obvious that the 2-timescale model continues to heat although radiative forcing stabilizes, and the 1-timescale model does not. Taken these results together, I do not see a (novel) advance here although the Discussion and presentation of the results is very well done. Overall a lot of times 'may' and 'might' are used indicating that this is more speculative. However, I hope that the authors can convince me that I am wrong. If not, I would propose to make it a review or perspective paper.*

Thanks for this point. We, in fact, fully agree that the analytical aspects of this paper do not represent sufficient quantitative advances in the understanding of combining constraints, and are rather intended to illustrate the wider conceptual points which are the focus of the paper (which the reviewer clearly appreciates, given the comments which follow). Given this (and similar assessments from the other reviewers), we have requested for the paper to be be classed as a review. In this context, we have removed the bootstrapping analysis - given the point has been largely covered by Caldwell 2018, Schlund 2020 and Brient 2019.

We see the paper's novelty as a conceptual framework to assess how emergent constraints relate to model assumptions. Given this, the toy model analysis does not provide deep insight on the nature of ocean timescale response, nor was it intended to. The point illustrates that if a model ensemble contains common approximations and a small number of degrees of freedom (in this case, represented by a single layer ocean), then strong emergent constraints can emerge which would confidently constrain the future response to a certain value - but the constraint may be broken, or simply not exist if a more complex model with additional degrees of freedom is used (here illustrated by a 2 layer ocean). The actual models used are incidental - they simply illustrate models at two different complexity levels. We go in in the discussion section to consider cases where such common oversimplifications may exist in the CMIP class models (e.g. soil respiration-temperature relationships).

*2) Many points that are discussed here as shortcomings of emergent constraints should, in my opinion, be attributed to the models themselves. Emergent constraints can, by definition, only improve the model output. If the models have structural shortcomings, they exist in the multi-model mean and the constrained results. Emergent constraints can thus 'only' improve the existing model output and cannot go further.*

We broadly agree that the key issue at hand is the structural shortcomings of the models. However, we disagree that the ECs can 'only improve'

existing model output - because of the implied precision in the value of the true future response which arises from considering only the error implicit in the regression relationship. An emergent constraint, literally, proposes that an unknown quantity is constrained, unlike an ensemble multi-model mean which just a model estimate with no pretense of precision. Thus, the precision of the EC can potentially be over-stated through the lack of consideration of aspects of common, oversimplified structural assumptions which create strong ensemble relationships. As such, although the structural errors are ultimately features of the models themselves, the use of ensemble relationships which are themselves subject to model errors impacts directly on the 'added value' of emergent constraints.

*The exchangability argument (lines 66-76, lines 296-304, lines 412-420) is thus wrongly stated in my opinion. An emergent constraint can only say: models that tend to simulate a large (small) variable A at present (e.g. extent of the Hadley Cell) also simulate a large (small) increase in variable B. If, and only if, a mechanistic relationship can be given and proven to a sufficient level, one could conclude that if models had rightly (as in the observations) simulated variable A, the model result for variable B would be the following. The constraint can never overcome model shortcomings or biases that exist across the entire model ensemble and is not designed to do that. The exchangability argument would thus only hold if the constrained result would be considered the truth, which it is obviously not. It is just an improved projection, but still based on imperfect models.*

This point is well taken, but contains an implicit assumption that the use of an emergent constraint can *only improve* a projection - where we would argue that ECs have the potential to make an overconfident projection due to a consideration of only the subset of available model validation data which was used in the EC. Fundamentally - the structural errors have the capacity not just to impact the simulated values of A and B, but also the the relationship between A and B - perhaps creating relationships which would not be present in a superset of models. We have already seen that different emergent constraints can emerge from the same ensemble with differing conclusions (e.g. the fluctuation dissipation relationship of Cox et al 2019, and the atmospheric stability relationship of Sherwood 2015). Either of these constraints used alone would constrain ECS to low or high values respectively. These problems are not insurmountable - the Bayesian framework of Williamson 2019 allow for the combination of different lines of evidence, but enacting such a strategy requires the modeling community to engage with (1) the independence of different constraints (e.g. Caldwell 2018) but also (2) the degree to which common structural errors could bias A, B or the relationship between them.

Our position is thus not that the constraints have no value, rather that the common assumption that a constraint used in isolation to constrain a projected ensemble distribution is not justified without considering other relevant aspects of model performance (including other ECs) and the realism of potential common structural assumptions which may have biased or created the emergent relationship. As we argue in our conclusions - we believe that there is a middle ground between standalone emergent constraints, and generic model multi-variate skill scores which could allow a focus on variables which might be of relevance to projected climate without relying exclusively on a single relationship.

*As an example, the present temperature and the leaf area index are within the current model ensemble good predictors of future GPP (Schlund et al., 2020). By using the present-day temperature and the leaf index, one can thus show how a model of the same ensemble would likely simulate future GPP if it had the right temperature and leaf index. Nevertheless, if the whole ensemble would be systematically biased and missing out an important process, like nitrogen limitation (lines 296-304), the emergent constraint cannot assess this. The systematic bias is present in the multi-model mean results and the constraint results and thus not primarily an issue of the constraint but the model ensemble. However, it is equally important to mention and assess these uncertainties when presenting mutli-model means and constrained results.*

We agree that the structural biases will impact the unconstrained model projection distribution, including the multi-model mean. But there are two further points to consider: (1) the multi-model mean and model distribution is not advertised as a calibrated estimate of a specific value, together with an implicit error and (2) the added value of the emergent constraint is also potentially subject to structural error.

The reviewer's example considers one example of structural error (the omission of N fertilization), and indeed - this is a case where the structural omission may create an over-confident constrained result if used to calibrate future GPP projections (if present day LAI simulation is a function of N-fertilization). We would argue that this is more problematic for the EC than the multi-model mean. Using the EC to calibrate future GPP is explicitly calibrating GPP to compensate for a missing process. This is an additional source of error because the effect of the bias of the missing process may be different for present day and future GPP, and therefore the constrained result stands to be confident, yet wrong.

The use of a single metric therefore throws away any information which is not in the constraint itself. In the reviewer's example, a model tuned to have the right temperature and LAI may be subject to other biases (in present day GPP or latent heat flux, for example), which would have been considered by the developers calibrating the model. As such, the ensemble distribution of projections and ensemble mean result are of course also subject to biases from missing process, but the models are expert tuned considering a wide range of metrics and therefore, to some degree more robust. The use of the EC to

calibrate projections effectively allocates zero weight to any orthogonal aspects of model performance (some of which may themselves be emergent constraints on the same quantity in the ensemble).

To put it in Bayesian terms, the prior ensemble distribution is subject to structural errors, but so are the relationships which provide the basis for the emergent constraint. Therefore, the ECs have the potential to be additionally impacted by structural errors than the unconstrained model distribution.

*Overall, I think the exchangability argument is wrongly stated as the models in an emergent constraint were never meant to be exchangeable, but observations are only used to inform the likely best guess of a given model ensemble. Having said this, the question remains how systematic model biases should be accounted for in uncertainty range. When calculating the multi-model mean, the standard deviation is often a measure of uncertainty, but this is not taking into account biases or structural errors. So, one can argue that the given uncertainties for a constrained result are equivalent to the uncertainties calculated by the multi-model standard deviation.*

We agree with the reviewer that the models were not intended to be exchangeable with reality. Model developers are generally keenly aware of the approximations made in their parameterizations. However, we maintain that the exchangeability assumption is implicit in the use of an emergent constraint to estimate the most likely value and uncertainty in that value. This is the view shared by Williamson (2019).

This problem does not apply to the simple estimate of mean and variance of the original model ensemble projections - which may be biased due to missing processes, but they are not, themselves, uncertainty estimates in a projected quantity. ECs, however, frequently use the error in the regression relationship to estimate an uncertainty in the projected quantity (see Cox 2018, Varney 2019 amongst many other examples) - and this additional step is a strong assumption, that the relationship is equally applicable to reality as it is to the models in the ensemble. We have expanded on this point when first introducing the concept of exchangeability in the introduction (line 94).

*Lines 412-420 make the point very clear. The current knowledge of the Earth System is as good as possible implemented in the Earth system models. Some processes known to be missing or wrongly represented, others are probably missing but not known yet (like the ocean mixing in the 1-timescale model). However, these missing processes are strictly speaking a problem of the models. If we had no knowledge of the ocean's importance (1-timescale model) and only lambda would be of importance, we had implemented no ocean in our models. Thus, the relationship would resemble the red points in figure 2e,f with model lambdas being different because of different atmospheric model components. By applying the hypothetically 'observed' lambda, we would reduce uncertainties re-*

*lated to the atmospheric model. We would, however, not find the right results because all models are missing the ocean component. Nevertheless, given the assumed hypothetical current state of knowledge (ocean is not important), and the consequent hypothetical models (no ocean), our knowledge of lambda and the emergent constraint would still 'improve' model projections under this assumption.*

Agreed, this broadly represents our intended illustration with the simple model - that the 'improvement' in this case would give a confident, but ultimately incorrect, constraint on the future dynamics of the model.

*Following this argument in its strict sense, would lead to the conclusion that models cannot be used because important steps might be missing and NOT that emergent constraints cannot be used. The 'assessment of underlying model assumptions' (line 436) should always be done if model output of any type is published, constrained or not constrained. If I understand the conclusion in lines 676-678 right, the authors argue that CMIP models are a comprehensive representation of the Earth System. Following that line of argument, we should not use them either to make projections of climate change at all? Or under which conditions?*

Here we differ from the reviewer. The single layer ocean is useful for some applications. It can capture some of the first order dynamics of the climate system given appropriate parameters. But the nature of that calibration matters. The Emergent Constraint approach takes an extreme position - that only one aspect of the model should be used in tuning (that which is correlated with the future response). This results in a very tight constraint on the single layer model's parameter space, but because the relationship itself is biased by the structural errors in the model - the constrained value is incorrect.

A more robust calibration strategy is to use all available relevant data to calibrate the model (e.g. the entire historical timeseries, paleoclimate records etc). In this case, individual aspects of the model error would still be subject to structural deficiencies, but trade-offs between tuning to different observations would reduce the degree to which the parameters are constrained (see Sanderson 2008 for an example of this in ESMs).

In short, we agree with the reviewer that all model outputs are subject to structural error. But we argue that the sole reliance on projection calibration from a single ensemble relationship in ECs introduces a particularly acute exposure to specific aspects of model error which will invariably lead to overconfident constrained projections.

*3) The manuscripts title, abstract, Conclusion (and partly Introduction) suggests that emergent constraints across the field of Earth Science are addressed. However, throughout the manuscript it becomes clear that the focus*

*of the manuscript is on emergent constraints of the Equilibrium Climate Sensitivity. In addition, other constraints, such as for the Transient Climate Response and the land carbon cycle, are (briefly) discussed. However, constraints for the ocean, a major part of the Earth System, are not discussed at all. Thus, the title and abstract are highly misleading. I would suggest to either clearly indicate that the manuscript is on emergent constraints on ECS and only discuss ECS constraints or add examples of ocean constraints and largely expand their exposition in section 5. Examples for emergent constraints in oceanography would be: Kessler et al. (2016), Kiwatkowski et al. (2017), Goris et al. (2018), Terhaar et al. (2020a), Terhaar et al. (2020b). The list is very likely not sufficient.*

This is a fair point. Our objective is certainly to talk about the methodology of emergent constraints in general - the key conceptual arguments are not specifically associated with climate sensitivity. We do not seek to exhaustively review or list every published emergent constraint here (this has been done elsewhere, e.g. in Brient 2020, Hall 2019 or Williamson 2018). But - the reviewer is correct that an ocean-specific example would be desirable to illustrate relevant structural assumptions for different broad genres of constraint. We have added a new section 5.3 on constraints on future ocean carbon uptake. Many thanks for the references.

*Furthermore, the ocean is ignored in questions about the ECS or atmospheric $CO_2$ (point 6), although the 2 timescale models clearly indicate that the ocean is important.*

Point well taken. We have revised the case studies to include ocean processes at relevant points.

*4) I am not a statistician, but I have strong concerns regarding the application of statistics in this study. First, I disagree that the Sherwood "D" and Cox constraints are correlated (Line 141). A r (if it is r) of 0.31 is a r2 of less than 0.1. A p-value is not given but I do not expect it to be supportive of a correlation. Even for Lipat and Qu, the r2 is 'only' 0.33. Please do not use the term correlated if the variables are not statistically correlated. Second, the two constrained ECS do not disagree (Line 142). Sherwood et al. (2014) find an ECS likely at 4°C with 3°C as a lower limit. Cox et al. (2018) report 2.8 ± 0.6 °C. Within the uncertainty ranges, they agree with each other. Lipat et al. (2018) and Qu et al. (2014) do not even give a constrained result for ECS as far as I can see this. But for this argument we can use the Schlund et al. (2020) estimate for the EC from Lipat et al., which is 3.0 ± 0.8 °C and try to read the result for Qu et a. (2014) from the corresponding subpanel, leading to 3.5 ± 0.4°C. These two constrains do also agree. The following paragraph paragraph and conclusions are thus wrong.*

The revised paper has removed this section entirely - given the paper was already long, and the assessment of EC correlation has been well discussed

by Caldwell (2018)

*5) The authors often use the argument that emergent constraints might be confusing to policy makers or other people. Furthermore, they say speak about their 'literal interpretation (line 196). I can see no evidence supporting this claim. On the contrary most emergent constraints only give a 'likely' estimate (summarized in table 4 in Schlund et al. (2020)) and even if all ECSs were used to give a best estimate with an uncertainty range, all ECS would agree. Thus, these ECS seem to be used to exclude outliers and not give a narrowly constrained result. Given that they all agree, I do not see the possible confusion.*

Thanks for this point. We have removed the specific references to climate policymaking. We would counter, however, that the majority of emergent constraints use probabilistic language in their primary conclusions - but in almost all cases, these probabilities exclude the potential for errors which are the focus of this study (that is, the uncertainty arising from model common simplifications which project onto either simulated quantities or intra-ensemble relationships).

*6) Lines 442-446: The authors claim that the differences in atmospheric $CO_2$ are caused by the land carbon sink, whereas Hoffmann et al. (2014) clearly state that "Weak ocean carbon uptake in many ESMs contributed to this bias, based on comparisons with observations of ocean and atmospheric anthropogenic carbon inventories." While the land carbon sink is very uncertain, the ocean has been found by Hoffmann et al (2014) to cause the bias. Please correct your paragraph accordingly.*

Totally agreed. We apologise for the land-centric discussion and have updated the paragraph accordingly.

*I would also argue that the bias-persistence in the too small ocean carbon sink (Kessler et al. (2016), Goris et al. (2018)) is caused by the circulation differences and is persistent over large timescales and thus not overconfident. The whole section should hence be replaced.*

We agree with the reviewer that ocean circulation biases play a significant role. However, we disagree that this is the sole source of uncertainty in atmospheric $CO_2$ concentration biases. As we discuss in the revised section - this is a trade-off between numerous factors: ocean productivity, circulation, land carbon and concentration feedbacks and soil temperature response. Systematic common biases in any of these aspects would qualify as a potential structural uncertainty in future $CO_2$ concentrations.

**2   Minor comments:**

*1) Figure 1: The panels are too small and impossible to read, especially on the diagonal. I suggest keeping y and x labels with the name of the constraint only at the left column and the bottom line. Furthermore, I cannot understand the added value from the bootstrapping algorithm from the manuscript. Often the uncertainty of the fit is estimated by prediction intervals (Bracegirdle et al. 2012; Nijsse et al. 2020; ...). To which degree and why does the bootstrap method improve the results, or the estimated uncertainties compared to these prediction intervals. If no improvement exist, why would you not just show the published results (Schlund et al. 2020)? And if you recompute them, why not showing the mean estimate + uncertainties + r2 or something similar. At the moment the subpanels do not allow to assess the mean, the uncertainty or anything else because they are too small.*

We have removed this figure in the revised version.

*2) Line 102: Table 3 is mentioned in the text before table 2.*

Table 1,3 now removed. Labelling fixed.

*3) Is the bootstrap approach general knowledge? If not, please consider telling the reader how it works or give a reference.*

Analysis removed in revision.

*4) Lines 114-121: This is very hard to read, and I am not sure that I understand the message. Could you try to rephrase it and make sure that the reader understands when a combination is appropriate and when not and why?*

This section has been removed in revision

*5) In general: How do you define correlation (Pearson's product-moment coefficient r or something else?)*

This section has been removed in revision

*6) Lines 206 to 209: I do not agree with this statement. Let's assume variable A (present) is correlated to variable B (projection) across a model ensemble and the correlation is mechanistically profound and supported by theory and observations. If variable A is now a very complex interplay of many processes, it could have a large inter-model spread without a lack of diversity. Thus, the presence of an EC can be a lack of diversity or a complex interplay of different processes. The sentence now is rather misleading.*

Agreed. and again, this section has now been removed.

*7) Lines 227-230: You should include Kessler et al (2016) and Goris et al. (2018) here.*

Thanks - agreed. Added.

*8) Lines 250-267: You should add Terhaar et al. (2020a,b) here, although it is not strictly a feedback process but identifying the leading order process that describes the future response.*

Thanks - we agree that these are process-based constraints, we've removed the word "feedback" from the title to allow a broader definition.

*9) Lines 269-285: You should add Kwiatkowski et al. (2017) here.*

Agreed, thanks.

*10) Line 288-290: I again do not agree with this sentence. A set of models with very complicated assumptions in different processes that govern both related variables, variable A (observable) and variable B (projected), would lead to a large spread in A and B and possible to a good correlation and EC.*

We disagree with this. If there are a small number of parameters governing a process in similar parameterisation schemes throughout the ensemble - the response of such models to both historical and future forcings will be a function of that small set of parameters, increasing the chance that an emergent relationship might be found.

Soil respiration/temperature relationships are a good example in CMIP, as we point out in the following paragraph. A simple temperature dependency equation has good skill in representing soil respiration in CMIP as a function of temperatures (Todd-Brown et al. 2013), and this enables strong emergent constraints on future soil respiration temperature sensitivity such as Varney (2020). But - this simple temperature relationship also fails to represent a large fraction of spatial variability in observed soil respiration (Todd-Brown et al. 2013) - which is potentially attributable to the common over-simplicity of the representation of the process in the ensemble.

Adding complexity in this case may indeed increase variance in the predictor or predictand, but the increased number of degrees of freedom in the process representation has the potential to add noise to the relationship between the two. However, if the ensemble diversity can be demonstrably reduced to a single process equation with (in the extreme case) one free parameter - responses to different forcings will be correlated by construction.

*11) Equations are not numbered*

Fixed

*12) Equation on line 341 is difficult to read (latex problem?)*

Thanks, fixed

*13) Lines 492-499: What is the added information here? It sounds more speculative than informative.*

We've deleted this paragraph.

*14) Section 5.3: Your two timescale models are constructed to make just this point. Maybe you could use this here and emphasize hence the importance of the ocean for long-scale warming (ECS) and point out that the difference in the ocean may, according to your model, be responsible for the different long-term temperature trajectories.*

Thanks for this suggestion - we've incorporated this discussion as suggested.

*15) Lines 537-644: I do not see from which results this conclusion is drawn. Could you pleas just point me to it? And what other metrics are you referring to here?*

We've deleted this paragraph - as the core of the argument is more clearly repeated in the following paragraph.

*16) Lines 554-660: Please cite here the multi-variable approach by Schlund et al. (2020)*

Thanks - yes. There's also a valid point from this paper on the significance of constraints persisting for multiple generations.

**3 References**

Williamson, D. B., Sansom, P. G. (2019). How are emergent constraints quantifying uncertainty and what do they leave behind?. Bulletin of the American Meteorological Society, 100(12), 2571-2588.

Varney, R. M., Chadburn, S. E., Friedlingstein, P., Burke, E. J., Koven, C. D., Hugelius, G., Cox, P. M. (2020). A spatial emergent constraint on the sensitivity of soil carbon turnover to global warming. Nature communications, 11(1), 1-8.

Cox, Peter M., Chris Huntingford, and Mark S. Williamson. "Emergent constraint on equilibrium climate sensitivity from global temperature variability." Nature 553.7688 (2018): 319-322.

Sanderson, B. M., Knutti, R., Aina, T., Christensen, C., Faull, N., Frame, D. J., ... Allen, M. R. (2008). Constraints on model response to greenhouse gas forcing and the role of subgrid-scale processes. Journal of Climate, 21(11), 2384-2400.

Brient, Florent. "Reducing uncertainties in climate projections with emergent constraints: Concepts, examples and prospects." Advances in Atmospheric Sciences 37, no. 1 (2020): 1-15.

Todd-Brown, K. E. O., Randerson, J. T., Post, W. M., Hoffman, F. M., Tarnocai, C., Schuur, E. A. G., Allison, S. D. (2013). Causes of variation in soil carbon simulations from CMIP5 Earth system models and comparison with observations. Biogeosciences, 10(3), 1717-1736.

**On structural errors in emergent constraints**

**Response to reviewer 2**

**May 25, 2021**

*Sanderson et al provide discuss the nature of emergent constraints (ECs), particularly the potential role of structural errors in driving uncertaints in ECs. Overall, I found it difficult to know what to do with this paper. While the material is clearly presented and interesting to read, it feels more like a review article or a perspective, rather than a journal article. The discussion is generally qualitative and/or speculative, and I'm still not quite sure what the main takeaways are.*

Thanks to the reviewer for the feedback on the manuscript. The opinion that the paper is better categorised as a review or perspective was shared by the other reviewers, and the paper is now classed as a review by the journal (and we are framing it as such).

This paper was never intended to be a quantitative analysis - rather as a perspective on the application of emergent constraints found in an ensemble of model simulations where we know there exist structural limitations to models which are represented in model simulations.

Our takeaways are that we demonstrate in a simple model ensembles how structural errors can produce overconfident emergent constraints, introduce a classification scheme for emergent constraints, and discuss a large number of case studies on where structural limitations in current models may be an additional source of error in published constraints.

Furthermore, we would argue that this general qualitative discussion is necessary in the literature before specific quantitative advances can be made. The nature of structural errors, by definition, is processes which are not resolved or incorrectly resolved in the model. As such, a quantitative analysis of such errors is not necessarily feasible, but this does not mean that this source of error can be completely disregarded when considering constrained distributions of projected climate quantities produced through consideration of ensemble derived correlations - as is the case in the majority of EC studies published to date.

*The quantitative analysis in the paper is limited to Figure 1, which shows the relationships between a number of previously published ECs ... covered in more detail in papers led by Caldwell, Bretherton and Schlund*

Given the classification of the paper as a review, we have removed Figure 1 entirely - which originally served to illustrate inter-constraint relationships, but we agree that this topic is adequately covered by Caldwell, Bretherton and Schlund.

*Figure 2, which uses the simple energy balance models to illustrate different kinds of ECs identified in the paper. The 2-layer energy balance model has been extensively discussed by Geoffroy et al, Armour, and Lutsko  Popp (some of these papers are cited in the present manuscript).*

We agree that Geoffroy, Armour and Lutsko have produced fine papers illustrating the dynamical assumptions of simple energy balance models, but they do not make the point we are making here - that an overly simple model structure can produce very strong emergent constraints which can be demonstrated to be overconfident in the context of information provided by a more complex class of models.

*Without more novelty or substance, it is difficult to recommend publication, although I enjoyed reading the manuscript.*

We hope our arguments here persuade the reviewer of the novelty of the article - and the need for a non-quantitative discussion on this topic. We hope further that the restructuring of the manuscript as a review will better reflect the article's purpose.

*Moving forward, the authors might want to think of ways to deepen their analysis. One approach might be to develop a mathematical framework or procedure for identifying and speaking about structural errors in emergent constraints.*

We believe this paper has already been written, by Williamson (2019) - which motivated us to write the present study. Williamson (2019) demonstrated how structural errors could theoretically be implemented in regression-based constraints in a Bayesian statistical model, but did not discuss any concrete case studies of what such structural errors would look like, In this study, we highlight potential sources of structural error and potential paths forward to future approaches which could better represent these errors in constraints.

*Alternatively, they could focus in on a particular kind of emergent constraint and probe the structural assumptions used by this kind of emergent constraint in more depth. For example, they could dig into the cloud schemes responsible for the process-based constraints on ECS (e.g., the Sherwood, Brient and Zhai constraints) to really understand the underlying structures. A template*

*could be the recent paper by Thackeray et al, which investigates the snow albedo feedback over multiple generations of climate models, including its relationship to the well established emergent constraint on the feedback.*

Papers focusing on the structural assumptions in individual constraints are enormously valuable, and our section 5 discusses a large number of case studies and how sources of structural error might arise. However, to objectively quantify these errors for any of the studies referenced would be a study in itself, and is far beyond the scope of this article.

In conclusion, we see the necessity for a general article on the potential for structural errors in emergent constraints, rather than a deep dive into a specific process, because there is a general lack of discussion of such errors in papers published to date - with a near blanket assumption that relationships found between predictors and predictands in the multi-model ensemble can be used to reduce uncertainty in projected quantities. In the study, we have demonstrated a simple model case where this assumption is demonstrably false, and we have highlighted where relevant structural assumptions may exist in the current model archive. We hope, in turn, that this will provoke future study which will work towards building constraints which are more robust to structural errors.

Technical Corrections:

*-The title is vague: "On Structural Errors in Emergent Constraints", and again makes it hard to know what the main takeaways are.*

Revised to "On the potential for structural errors in emergent constraints" - which well captures our topic.

*-In the first sentence of the introduction, I'm not sure it's right to state that higher CO2 concentrations are a "boundary condition which has yet to be realized". Increasing CO2 concentrations doesn't enter the boundary conditions, it adds a forcing term. So I would describe climate forecasting as an initial value problem, rather than a boundary condition problem.*

We have changed "boundary conditions" to "forcings" to remove this ambiguity.

More generally - we agree, that if humans are considered part of the Earth System, then it could be argued that climate projections are an initial condition problem (with boundary conditions being only variations in solar forcing). However, it is generally accepted in the climate literature that the initial conditions refer only to the state of physical climate system at the time of initialization (see, e.g. Hawkins and Sutton 2009, Deser 2012, Hawkins 2016, Sriver 2015).

*-The paper claims that the Cox and Sherwood D constraints are well correlated with each other. But in fact the correlation co-efficient is only 0.31 ( 10pct of the variance explained).*

This section is entirely deleted in the new version - but we do agree that this sentence was incorrect.

*-Typo at L610: "wider constrained range wider"*

Thanks, fixed.

*-L654: In terms of multi-metric approches, the authors may wish to cite the "cloud-controlling factor" approach (see Klein et al for a recent review) which has recently shown promise for constraining cloud feedbacks.*

Thanks for this. Completely agreed that there is an argument for greater robustness through the consideration of "bottom-up" decomposition of net feedbacks, such as that demonstrated in the Klein paper. We've added a paragraph on this topic in the discussion.

**1 References:**

Williamson, D. B., Sansom, P. G. (2019). How are emergent constraints quantifying uncertainty and what do they leave behind?. Bulletin of the American Meteorological Society, 100(12), 2571-2588.

Klein S.A., Hall A., Norris J.R., Pincus R. (2017) Low-Cloud Feedbacks from Cloud-Controlling Factors: A Review. In: Pincus R., Winker D., Bony S., Stevens B. (eds) Shallow Clouds, Water Vapor, Circulation, and Climate Sensitivity. Space Sciences Series of ISSI, vol 65. Springer, Cham.

Hawkins, E., Sutton, R. (2009). The potential to narrow uncertainty in regional climate predictions. Bulletin of the American Meteorological Society, 90(8), 1095-1108.

Deser, C., Phillips, A., Bourdette, V., Teng, H. (2012). Uncertainty in climate change projections: the role of internal variability. Climate dynamics, 38(3), 527-546.

Hawkins, E., Smith, R. S., Gregory, J. M., Stainforth, D. A. (2016). Irreducible uncertainty in near-term climate projections. Climate Dynamics, 46(11), 3807-3819.

Sriver, R. L., Forest, C. E., Keller, K. (2015). Effects of initial conditions uncertainty on regional climate variability: An analysis using a lowresolution CESM ensemble. Geophysical Research Letters, 42(13), 5468-5476.

Armour KC (2017) Energy budget constraints on climate sensitivity in light of inconstant climate feedbacks, Nature Climate Change, 7, 331-335

Bretherton, C. and Caldwell, P.: Combining Emergent Constraints for Climate Sensitivity, Journal of Climate, 33(17), 7413–7430. 2020

Caldwell, P. M., Bretherton, C. S., Zelinka, M. D., Klein, S. A., Santer, B. D. and Sanderson, B. M.: Statistical significance of climate sensitivity predictors obtained by data mining, Geophysical Research Letters, 41(5), 1803–1808, doi:10.1002/2014gl059205, 2014.

Caldwell, P. M., Zelinka, M. D. and Klein, S. A.: Evaluating Emergent Constraints on Equilibrium Climate Sensitivity, Journal of Climate, 31(10), 3921–3942. 2018

Geoffroy, O., Saint-Martin, D., Olivié, D. J. L., Voldoire, A., Bellon, G. and Tytéca, S.: Transient Climate Response in a Two-Layer Energy-Balance Model. Part I: Analytical Solution and Parameter Calibration Using CMIP5 AOGCM Experiments, J. Clim., 26(6), 1841–1857, 2013a.

Geoffroy, O., Saint-Martin, D., Bellon, G., Voldoire, A., Olivié, D. J. L. and Tytéca, S.: Transient Climate Response in a Two-Layer Energy-Balance Model. Part II: Representation of the Efficacy of Deep-Ocean Heat Uptake and Validation for CMIP5 AOGCMs, Journal of Climate, 26(6), 1859–1876. 2013b.

Lutsko, N. J., Popp, M. (2019). Probing the sources of uncertainty in transient warming on different timescales. Geophysical Research Letters, 46, 11367– 11377

Schlund, M., Lauer, A., Gentine, P., Sherwood S. C. and Eyring, V. (2020) Emergent constraints on equilibrium climate sensitivity in CMIP5: do they hold for CMIP6? Earth Syst. Dynam., 11, 1233–1258.

**On structural errors in emergent constraints**

Response to reviewer 3

May 25, 2021

*The authors highlight the disagreement present between different estimates of the same quantity arising from multiple single metric emergent constraints and the confusion this may cause. They claim that the cause of this disagreement is over-simplified process representation and a lack of diversity among climate models, and illustrate their argument using a class of simple climate models.*

Agreed. Many thanks to the reviewer for the comments and careful review.

*Overall I find little to criticise, except that the manuscript reads more like a review than an original research paper.*

This opinion was reflected in all the reviews (and the authors broadly agree). Given this, we requested that the journal reclassify the paper and we new present the study as a review.

*The discussion of the different classes of emergent constraints largely reflects what has been written elsewhere (and cited within the manuscript. The main original contribution appears to be the illustrative example, which is informative and welcome. The generality of the conclusions from such an extreme example is perhaps questionable, but many processes within climate models are parametrised equally simply, or more so.*

Thanks for the comments. We agree that the simple model is an extreme example of a structural error, which cannot be generalised to parametric limitations in CMIP-class models. However, we do think that the simple example serves to illustrate the potential for how structural errors could project onto emergent constraints, and ultimately to encourage future studies to consider how ensemble-derived relationships could, in fact, arise from simple common parametric assumptions which exist throughout the ensemble. We have endeavored in the revisions to sharpen our language in making the distinction between the toy model and the discussion of CMIP structural errors.

**1   Minor points:**

*Lines 17-18: From a purely statistical perspective there is no contradiction here. Inferences about the same quantity derived from different lines of evidence are not expected to be the same. As argued by Williamson and Sansom (2019), it is likely that many of these estimates are over-confident. But even if a more appropriate uncertainty quantification were carried out, the inferences would not and should not be the same. However, I accept that this is confusing for policy makers.*

Agreed, apologies for the lazy wording - we've revised as:

"The prevalence of this thinking has led to literature which made confident, yet inconsistent between studies, claims on the probability bounds of key climate variables. "

In general, we agree that there is the potential to use multiple ECs and combine lines of independent evidence. However, we would argue that the majority of EC studies do not present their results as likelihoods which could inform a combined posterior or body of evidence - rather they present their headline result as an absolute probability (see, e.g. Cox 2018, Nijsse 2020, Jiménez-de-la-Cuesta 2019).

*Line 19: More likely than what? This is a leading statement, the authors only show that this is one possible explanation for emergent behaviour*

Agreed. We've revised the sentence as follows:

"Here, we illustrate that strong ensemble relationships between observables and future climate can arise from common structural assumptions with few degrees of freedom. Such cases have the potential to produce strong, yet overconfident constraints when processes are represented in a common, oversimplified fashion throughout the ensemble, where we might have low confidence in the behaviour of the process in a future climate. "

*Lines 66-67: A constraint disappearing in later generations of climate models is not necessarily proof that the constraint was spurious. Convergence among models may make the spread among the observable and/or the target quantity to small for a "significant" relationship to be detectable, the relationship may even appear to change sign. Although, convergence has its own caveats if it is not due to advances in knowledge but rather common acceptance of a least bad solution.*

We agree - the original sentence was sloppily worded. We intended to convey the case (as in Klein and Hall 2015), where the new models are outliers in the original constraint, thus degrading confidence in the relationship when

the two ensembles are combined. We've revised the sentence to clarify.

*Lines 93-95: The pseudo-Bayesian approaches cited have very limited applicability, and most principled statistical approaches to the analysis of emergent constraints rely on regression analysis which does not imply model weights.*

Agreed, thanks. Added the caveat that these studies do not test the underlying implicit assumptions of the regression framework.

*Line 98: Some level of subjectivity is unavoidable, the idea of objectivity in science where data are interpreted through any model is a delusion. The resampling approach used in the following paragraph applies avoids making parametric assumptions about probability distributions but implies certain assumptions of its own which aren't clearly stated.*

Agreed. The analysis has been removed in the revision, following comments from other reviewers, but we fully agree on the impossibility of truly objective analysis

*Lines 138-141: The correlation between the Cox and Sherwood D constraints is relatively week and unlikely to be statistically "significant".*

Agreed - apologies for the error. The relevant section has been deleted.

*Lines 215-218: See my previous comment on Lines 66-67.*

Again, the wording was sloppy. Now:

"If an emergent constraint has been found in an MME (providing it has not been demonstrated to be statistically spurious by, for example, additional models which significantly weaken the correlation (Klein and Hall 2015))..."

*Line 369: What is meant by a meaningful constraint? The two later model indicates a reduction in uncertainty.*

Sorry, reference error. The sentence (now reworded) refers to Figure 1d - which shows that T70 is not strongly correlated with T280 in the 2 layer model ensemble.

*Line 369: Figure 2d?* Yes, sorry - see above.

*Line 654: Karpechko et al. 2013 (DOI: 10.1175/JAS-D-13-071.1) should be cited here.*

Agreed - added.

**2  References**

Cox, P. M., Huntingford, C., Williamson, M. S. (2018). Emergent constraint on equilibrium climate sensitivity from global temperature variability. Nature, 553(7688), 319-322.

Nijsse, F. J. M. M., Cox, P. M., Williamson, M. S. (2020). An emergent constraint on Transient Climate Response from simulated historical warming in CMIP6 models. Earth System Dynamics Discussions, 2020, 1-14.

Jiménez-de-la-Cuesta, D., Mauritsen, T. (2019). Emergent constraints on Earth's transient and equilibrium response to doubled CO 2 from post-1970s global warming. Nature Geoscience, 12(11), 902-905.

---

## Author Response (AR2)

*The authors have gone to great length to take into account many comments of the four reviewers and have in that process substantially improved the manuscript. However, despite this great work, I feel that some of my major points have not been sufficiently addressed and strong disagreements remain. Most importantly, it reads to me still more like an opinion piece, a very interesting opinion piece indeed, but with very little evidence to support the claims made in the manuscript. As already said before, it is very well written and a pleasure to read. However, I would have liked if the proposition by reviewer 2 to look at some ECS in much more detail would have been taken into consideration. Now it still remains quite speculative.*

Thanks to the reviewer for taking the time to re-review the paper. We appreciate that this paper occupies an unusual niche for ESD – but the points made in the paper are not specific to climate sensitivity, they relate to the practise of making inference on an unknown climate parameter using correlations obtained from a structural model ensemble. Thus, although we refer in a number of places to examples which discuss climate sensitivity, this is not our focus.

That said, the reviewer raises a number of reasonable points which we have endeavoured to address better in the manuscript. The reviewer is correct that a clear line needs to be drawn between speculation and objective findings. For the simple model, the objective findings are very simple – that the presence and strength of a constraint are conditional on common structural model assumptions. Applying this logic to the complex models is more nuanced – and is dependent, as the reviewer suggests, on our absolute confidence in process representation in CMIP class models. The point is well taken that process biases ultimately exist in the models, and not in the constraints themselves. We attempt to clarify this in the revised version.

*1) I feel that the authors judgement of the three kinds of emergent constraints is strongly biased in favor of type 1. The constraint of type 1 seems to be given more confidence than the ones of type 2 and 3. I find little to no justification why this should be the case.*

*For example, the authors mention that the EC that relates past warming to TCR is likely robust. So far, I do not see why this should necessarily be the case. What if a tipping point occurs: freshening of the Southern Ocean shuts down deep convection and no warm subsurface waters comes to the surface and heat uptake would be altered? The very different sea ice extent in the CMIP5 and CMIP6 models could move this moment to the early 21st century or to the end of the 21st century or even into the 22nd century. As such, it would also have strong consequences for the TCR and historic warming. historic warming due to a change in albedo and cannot act as a factor later on?*

*An example is the NorESM model that has a huge Southern Ocean Sea ice mass, much larger than the other models but a rather normal extent, suggesting that a very thick part of sea ice exist. Whereas the first sea ice (thin) disappears quickly, the thick part takes centuries to disappear. Or what if Arctic Sea Ice was melted early in the model and resulted in a strong albedo change but in others this comes later, maybe even after the 70 years of the 1% run that is used to quantify TCR? Without a mechanistic explanation, historic vs future trend relationship could potentially be pure 'luck'. I would argue that a type 1 constraint is, without a mechanistic explanation of the underlying processes, much less robust than an EC that identifies the driving process.*

Thanks for this point and its illustration with a detailed example.  We have reassessed our statements about type 1 constraints.  We fully agree that the presence of a tipping point or mechanism for the transient trend to alter would weaken a first order constraint.  Indeed, we already note this when we introduce the concept.

We do agree, however, that our level of confidence on the TCR constraint may be too high in places, and we've edited section 5.4 to convey this.  However, we do not see evidence of deviation in either CMIP ensemble of near-linear warming response to linearly-increasing forcing.  See Gregory (2015), and the following plot, which shows in CMIP5 and CMIP6 there is a very strong relationship in both ensembles between warming at 400ppm (approx. present day) and warming and TCR at 560ppm (2xCO2) in the 1pctCO2 simulations, with a slightly weaker relationship for warming at 4xCO2:

This points to at least the potential for extrapolation of a forced transient trend in CMIP ensembles in the idealised case of the 4xCO2 simulation, potentially complicated by forcing uncertainties in a historical simulation.  However, the point is well taken that this is an empirical observation (we believe the constraint because it's there) – and there is strength in your argument that a process-based constraint allows for a deeper investigation for responses like TCR and ECS, which are driven by multiple mechanisms.  We have updated the text to reflect this better.

*What intrigues me in this example (Tokarska et al., 2020) even more is that the slope of the EC changes from CMIP5 to CMIP6 by around 100%. So, an overestimation of the historic warming by 0.1°C would have a twice as large effect on TCR. This suggest that other processes are in place and merits more discussion and that two relationships are found in both model ensembles but apparently not the same. This seems to remain undiscussed in that paper and in this review, which assesses other types of constraints much more critical. Having said all this, I find the claim that type 1 constraints are more robust unfounded.*

[Figure]

This was an interesting observation, and we briefly followed it up.  The plot above shows that the relationship between transient response on different timescales is strong in both CMIP5 and CMIP6 in the 1pctCO2 simulations.

More important than the slope itself is perhaps the fact that the relationship between past warming and TCR in CMIP5 showing in Tokarska 2020, is much weaker – with a 0.52 correlation – so confidence in the slope is lower than the CMIP6 case (corelated at 0.74).  We agree that the lack of strong relationship in CMIP5 is a relevant issue, especially given the addition of the CMIP5 to the CMIP6 ensemble weakens the constraint seen in CMIP6 alone.  We've adjusted the text to reflect this.

We would expect there to be two major factors in the difference between the two ensembles: CMIP6 containing some models with higher TCR values than CMIP5, and there are potentially differences in historical forcing trends between the two studies.  However, following this up further is a study in itself.

*2) On the other hand, type 2 constraints are more criticized although they identified the leading process. It states, "A plausible, robust, process-based EC is still conditional on the plausibility of the relevant process as it is represented in the class of models used in the ensemble."*

We stand by this sentence, but we take the reviewers point – that the plausibility of process representation can be reinforced by additional observations.  We've added the following qualifier:

**"However, confidence in process representation can be assessed and potentially increased through consideration of the plausibility of common model assumptions (Klein and Hall, 2015) or identification of independent observables which can be used to assess the degree to which models represent relevant processes (Terhaar et al., 2020)."**

*In many cases, these processes are demonstrated in observational studies. For example, Terhaar et al. (2020) have found their EC after observational studies indicated that deep water formation in the Barents Sea is responsible for most of the anthropogenic carbon inventory change in the Arctic Ocean. From my perspective, this is more robust: Identifying with observations the dominant process for a projection, see how this process is represented in models and see if the observational-based hypothesis holds in models. If this is the case, the EC should be considered very robust.*

We certainly agree that process understanding based on observations is highly desirable. However, process-based emergent constraints are still fundamentally based on the differences among model simulations, so in this sense models remain fundamental to the  interpretation of the constraints. Furthermore:

(1) this sequence of inquiry is not universal –it remains possible to retrofit a plausible process hypothesis upon the discovery of a constraint, and it is difficult to objectively assess from the published literature whether this has been done, given the primary quantitative evidence for an emergent constraint is generally presented as the constraint itself.

(2) Though the example you present of Terhaar (2020) is compelling (that the base-state observationally derived understanding of Arctic water transport provides a good conceptual model for an emergent constraint based on the persistence of ocean circulation), in this case, a simple model expectation (expected deep ocean carbon transport given persistence of circulation biases) is supported by the ESM relationships.  In this case, confidence in the EC is boosted by the persistence-based hypothesis, and the presence of the EC in the ESMs is

confirmation that nothing unexpected is happening in this class of model.  We would still argue that confidence in the EC is ultimately conditional on the ESM process representations being complete and accurate.

That said, we do agree that a strong hypothesis, with supporting observational evidence, allows the confidence to be built in process-based ECs through supporting, potentially independent means.

*3) This leads me to my main criticism, which was in my opinion not sufficiently addressed in the responses. The authors 'unload' model shortcomings almost entirely on the ECs and argue that model projections do not have uncertainties.*

*However, the IPCC report and multiple studies use the standard deviation across a model ensemble as uncertainty and the mean of the entire ensemble (or a subsample after excluding physically wrong models) as the best estimate. I hence strongly disagree with the response that MME do not make a statement about uncertainties. Like all scientific studies, the method of EC is not perfect and never claims. It, however, can help to analyse a model ensemble and to learn about its strengths and weaknesses.*

This point is well taken.  The use of the MME distribution mean and standard deviation in assessment is indeed making implicit assumptions about how the model distribution relates to uncertainties.  We agree that treating the CMIP distribution as a proxy for uncertainty is problematic – and this is also appreciated in the IPCC assessment process.  AR5 was quite specific with uncertainty language while regarding to the MME - and referred to the CMIP distribution as a range or spread, but generally not as an uncertainty. AR6 is also not going to take CMIP6 as an uncertainty because the distribution of CMIP6 ECS contains a significant fraction of models above the assessed likely range.

We also agree that ECs can be a powerful tool in identifying model feedback processes and relevant observables, and potentially for understanding ensemble limitations – and we have revised our discussion to highlight that these uses of ECs can be a powerful way to understand the ensemble.  Our critique is the use of ECs in isolation as a tool to confidently narrow projections while potentially ignoring other aspects of model performance.

*I think a fundamental misunderstanding between the authors and me is the way we interpret emergent constraints. To me emergent constraints help to analyze existing model outputs. To that extent they cannot erase strong shortcomings like missing processes (in most cases). They can however reduce uncertainties in the model ensemble because of how the existing knowledge is numerically represented (see lambda example). They hence reduce the uncertainties in existing projections and can account for an identified bias, but they may miss biases if all models do.*

Thanks for this point.  Firstly, we are in agreement with the reviewer that the biases and approximations ultimately exist in the models, and not in the constraints themselves.  We also agree that emergent constraints are potentially a powerful way to understand diversity in model results.

However, our primary point is that the use of emergent constraints to reduce model projection spread must be treated with caution because *if* models make common simplistic assumptions (as in some cases, we know they do – e.g. eddy diffusion parameterisations, soil temperature respiration relationships), then (1) an EC may emerge because there are few degrees of freedom in the common model structure repeated in the ensemble, (2) calibrating a projection using this EC is then conditional on those common

structural assumptions and (3) the EC framework disregards other observable quantities which might highlight the deficiencies of the model parameterisations.

We fully agree that ECs are useful to isolate potentially relevant observable processes for feedback processes in MMEs, but we differ on whether this information should then be used in isolation to constrain the MME distribution of projections. As we argue in the discussion – multi-metric skill scores and model weights represent one extreme (many metrics, no consideration of relationship to response), while emergent constraints represent the other (one metric chosen because of its correlation to response).

We argue that both approaches are non-ideal, and that the more defensible middle ground has been underexplored. Constraining projections based on an EC is a very strong statement that only the EC variable is relevant in our assessment of the plausibility of different values of the response variable, and all other model performance metrics can be ignored. However, it is only by consideration of multiple model metrics, and trade-offs between different calibration targets, that model structural errors become apparent (see e.g Hourdin 2017 or McNeall 2016). This multi-metric perspective highlights that complex models cannot be tuned to match all observable targets simultaneously, and by restricting our consideration to only one variable, we would get an overly confident projection of the future. Clearly – a simple skill score/bias weighting also has disadvantages, with no focus on aspects of model response which are relevant to the projected quantity.

As such, we agree with the reviewer that emergent constraints can help us analyse ensembles, and that they should inform which variables should be included in model evaluation. What we argue against is the use of emergent constraints as a direct means to constrain model projections, implicitly ignoring all other possible evaluation metrics as well as the known process assumptions in the component models.

*I will try to make my point clear with the simple example in the paper. First, the authors show that not assuming the deep ocean can lead to a wrong constraint on T280 warming by using the T70 warming. This is indeed the case; however, it is not an issue with the emergent constraint. The problem lies in the models that do not consider a deep ocean. If our knowledge does not include the deep ocean, we would expect the warming as simulated by model 1. The difference lies thus only in the difference of lambda, which itself would depend on how the feedback mechanisms are calculated. Within that model ensemble with very different lambdas, knowledge of the 'real' lambda or T70 would indeed improve the projection of that model ensemble. The EC would give the most likely projection assuming no deep ocean exist and hence improve the projection of such a model ensemble. This is, however, not reality, but that is due to the models and not the way these models are analyzed. The example in this manuscript is hence rather an example why model shortcomings are a problem and not emergent constraints.*

In the case of the simple model example, we agree that the bias exists in the shallow ocean model. However, in this simple case, we disagree that the EC derived in the simple model relating T70 to T280 would improve the projection. The use of that EC in that model would result in a constrained projection which excluded the real value of T280 (where 'real' is in this case the two-layer model). Use of the EC alone to constrain projections would therefore result in a failed forecast where the truth lies outside the constrained distribution. This failure happens because the EC does not factor in uncertainty due to the model structural errors.

In this case, the identification of the EC is still useful because it helps us understand the degrees of freedom in the model and the processes which govern its long-term response. It even helps us identify the model's structural limitations, given it illustrates that the ensemble does not represent a plausible diversity of equilibration responses. However, the example underlines that the use of the EC alone in this ensemble to constrain projections of long-term warming would result in a confident wrong answer – and our argument highlights the risk of this type of error.

*4) In the Conclusions, the authors argue that EC is effectively model weighting. I disagree. No model is weighted when using emergent constraints. An important mechanism is identified for a projection and that mechanism relates the predictor and predictand in the same way across all models, they are equally weighted.*

This is a misunderstanding - apologies, we have clarified our position. Our argument is not that weighting is used in the derivation of ECs, rather that their application to constrain projections is a form of weighting. We agree that almost all published ECs weight each model member equally in the derivation of the relationship between predictor and predictand. But ECs usually present a calibrated projection conditional on the observed value and uncertainty range of the predictor, usually using the ensemble as a transfer function. This is effectively weighting the projected values of the response according to modelled skill in the predictor.

*5) In general, I have the feeling, that the authors and me agree but that that the assessment of EC depends on the variable that is constrained. A local advection driven process, such as C uptake in the Southern Ocean, can by observations and models be linked to the formation of mode and intermediate waters. ECS or TCR are, however, depended on many different variables and unlikely to be constrained by one single process. I think this difference should be emphasized more strongly, especially given the Conclusions about multi-variable metrics. Overall, I feel that the ECs that constrain a local process are being taken prisoners by the often-spurious ECS constraints.*

Completely agreed – the composite response of a complex system is subject to a different set of considerations than a single process component, the latter enabling a clearer assessment of model assumptions and performance. We already discussed the difference between 'top-down' and 'bottom-up' emergent constraints in the discussion – we agree that Terhaar 2021 is an excellent example of the latter, and have included the citation.

*6) Could you add prediction intervals and r2 values on figure 1 please. That would help a lot.*

Agreed, done.

Hourdin, F., Mauritsen, T., Gettelman, A., Golaz, J. C., Balaji, V., Duan, Q., ... & Williamson, D. (2017). The art and science of climate model tuning. *Bulletin of the American Meteorological Society*, *98*(3), 589-602.

McNeall, D., Williams, J., Booth, B., Betts, R., Challenor, P., Wiltshire, A., & Sexton, D. (2016). The impact of structural error on parameter constraint in a climate model. *Earth System Dynamics*, *7*(4), 917-935.